# Kinesin-1 regulates antigen cross-presentation through the scission of tubulations from early endosomes in dendritic cells

Meriem Belabed[1], François-Xavier Mauvais [2], Sophia Maschalidi[1], Mathieu Kurowska[1], Nicolas Goudin[3], Jian-Dong Huang [4], Alain Fischer[1,5,6], Geneviève de Saint Basile[1], Peter van Endert [2], Fernando E. Sepulveda [1,7] & Gaël Ménasché [1✉]

Dendritic cells (DCs) constitute a specialized population of immune cells that present exogenous antigen (Ag) on major histocompatibility complex (MHC) class I molecules to initiate CD8 + T cell responses against pathogens and tumours. Although cross-presentation depends critically on the trafficking of Ag-containing intracellular vesicular compartments, the molecular machinery that regulates vesicular transport is incompletely understood. Here, we demonstrate that mice lacking Kif5b (the heavy chain of kinesin-1) in their DCs exhibit a major impairment in cross-presentation and thus a poor in vivo anti-tumour response. We find that kinesin-1 critically regulates antigen cross-presentation in DCs, by controlling Ag degradation, the endosomal pH, and MHC-I recycling. Mechanistically, kinesin-1 appears to regulate early endosome maturation by allowing the scission of endosomal tubulations. Our results highlight kinesin-1's role as a molecular checkpoint that modulates the balance between antigen degradation and cross-presentation.

[1] Université de Paris, Imagine Institute, Laboratory of Molecular basis of altered immune homeostasis, INSERM UMR1163, F-75015 Paris, France. [2] Université de Paris, INSERM, U1151, Institut Necker Enfants Malades; Université de Paris; CNRS, UMR8253, F-75015 Paris, France. [3] Cell Imaging Facility, Université de Paris, Imagine Institute, F-75015 Paris, France. [4] School of Biomedical Sciences, Li Ka Shing Faculty of Medicine, The University of Hong Kong, Hong Kong, China. [5] Immunology and Pediatric Hematology Department, Necker Children's Hospital, AP-HP, F-75015 Paris, France. [6] Collège de France, F-75005 Paris, France. [7] Centre national de la recherche scientifique (CNRS), F-75015 Paris, France. ✉email: gael.menasche@inserm.fr

Dendritic cells (DCs) constitute a specialised population of immune cells that excel in antigen (Ag) presentation and the initiation of adaptive immune responses. The cells process and present endogenous and exogenous Ags in association with class I or class II major histocompatibility complex (MHC) molecules, which respectively trigger CD8+ or CD4+ T-cell activation. DCs are particularly efficient, as compared to other antigen-presenting cells, for the presentation of exogenous Ags to CD8+ T cells, a process called cross-presentation[1,2]. Various subsets of conventional splenic DCs have been characterised; for example, the cDC1 (Xcr1+, CD8α+) subset is considered to be most proficient in cross-presentation[3]. However, other DC subsets (such as cDC2 (Xcr1−, CD11b+) also perform cross-presentation - albeit less proficiently[4]. During cross-presentation, extracellular Ags are internalised within specialised organelles termed phagosomes (for particulate Ags such as microbes or dead cells) or endosomes (for soluble Ags). Both compartments subsequently mature through fusion with late endosomes or lysosomes. This fusion leads to a slight acidification of organelles via the recruitment of V-ATPase and the NADPH oxidase 2 complex (NOX2), and the delivery of degrading enzymes that partially mediate Ag proteolysis[5,6]. This mild degradation allows Ags to be transported into the cytosol, where they are further degraded by the proteasome[7,8]. The resulting peptides are then either transported toward the endoplasmic reticulum (ER) or back into the endosomal/phagosomal compartment by TAP1/2 transporters[9]. The translocated peptides are trimmed by amino peptidases (in the ER) or by insulin-responsive amino peptidase (in the endosomal/phagosomal compartment) and are then loaded onto MHC-I molecules[10,11]. Lastly, the MHC-I-peptide complexes are transported to the plasma membrane to prime CD8+ T cell responses to pathogens, tumours and/or self-Ags. The Ag processing for the cross-presentation can also occur in a proteasome- and TAP-independent pathway known as the vacuolar pathway, in which Ag degradation requires the activity of acidic endosomal proteases and is followed by peptide loading onto MHC-I molecules within endosomes[12].

It has been suggested that internalised soluble Ags enter into distinct populations of early endosomes that differ with regard to their intracellular mobility and maturation kinetics[13]. One pathway matures rapidly and leads primarily to Ag presentation on MHC-II, whereas the second is more stable and favours MHC-I presentation. The endocytosis route depends on the mechanism of Ag uptake. The soluble ovalbumin (sOVA) model has been studied in detail. If internalised by the mannose receptor, sOVA is routed towards stable, early endosomes. In contrast, internalisation by pinocytosis targets sOVA to lysosomes[14]. The main difference between the two populations of early endosomes is that rapidly maturing endosomes move quickly along microtubules, whereas as slowly maturing endosomes are stationary[15].

In the present study, we focused on the conventional microtubule-dependent motor protein kinesin-1 (the archetypal member of the kinesin superfamily), which mediates cargo transport to the plus-end of microtubules. Kinesin-1 is a tetrameric protein constituted by two heavy chains (KIF5A, KIF5B, or KIF5C) and two light chains (KLC1, KLC2, KLC3 or KLC4)[16]. We have previously reported that the secretion of lysosome-related organelles (such as lytic granules in CTLs, secretory granules in mast cells, and alpha-granules and dense granules in platelets) is mediated by kinesin-1[17–19]. Apart from the kinesin-1-based transport of late endosomes/lysosomes towards their secretion site upon cell activation, few studies have looked at the role of this molecular motor in the regulation of early endosome and recycling endosome trafficking[20–22].

The precise molecular mechanisms regulating the processing of extracellular Ags have not been fully characterised. Here, we investigated the role of kinesin-1 in DC-mediated Ag cross-presentation by using *Kif5b^{fl/−} Vav1-Cre* conditional knockout (cKO^{Kif5b}) mice that lacked *Kif5b* in all their hematopoietic lineages (including DCs). Our results show that kinesin-1 (i) has an essential role in the Ag and MHC-I endocytic trafficking upstream of cross-presentation, and (ii) regulates early endosome movement and maturation via the fission of endosomal tubulations.

## Results

**Kinesin-1 deficiency impairs cross-presentation by DCs.** Given that trafficking of intracellular vesicular compartments is necessary for Ag cross-presentation, we assessed the role of the conventional microtubule-dependent motor protein kinesin-1 in Ag presentation by DCs. We generated the cKO^{Kif5b} mouse model, which lacks Kif5b expression in all hematopoietic cell lineages[18]. These mice display no obvious abnormal development of the lymphoid lineage (Supplementary Fig. 1). We confirmed using quantitative real-time PCR assays that Kif5b was absent in CD8α+ and CD11b+ DCs purified from the spleen of cKO^{Kif5b} mice and in bone marrow-derived DCs (BMDCs), and we did not observe compensatory up-regulation of the other isoforms (Kif5a and/or Kif5c) (Fig. 1a). Despite the absence of Kif5b, conventional DCs developed normally in cKO^{Kif5b} mice (Fig. 1b, c). Bone marrow progenitors differentiated normally into BMDCs, and responded normally to lipopolysaccharide (Supplementary Fig. 2). CD8α+ and CD11b+ DCs purified from the spleen of wild-type (WT) and cKO^{Kif5b} mice were tested for their ability to cross-present sOVA to transgenic OVA-specific (OT-I) T-cell receptor (TCR) CD8+ T cells in vitro (Supplementary Fig. 3a). CD8α+ and CD11b+ DCs from WT mice cross-presented sOVA in a dose-dependent manner; however, CD8α+ and CD11b+ DCs from cKO^{Kif5b} mice induced significantly less interleukin 2 (IL-2) secretion from OT-I T cells at the highest tested concentrations of sOVA (Fig. 1d, e). Likewise, Kif5b-deficient BMDCs were strongly impaired in their ability to cross-present sOVA, relative to WT BMDCs (Fig. 1f). In a control experiment, CD8α+ and CD11b+ DCs and BMDCs from WT vs. cKO^{Kif5b} did not differ in their ability to present the OVA epitope S8L (SIINFEKL peptide, OVA_{257-64}) (Fig. 1g). These results suggest that the impairment in cross-presentation of DCs in the absence of Kif5b was not related to impaired expression of MHC-I at the plasma membrane. In order to assess kinesin-1's role in particulate Ag presentation, we studied the cross-presentation of OVA coupled to latex beads. A similar defect in cross-presentation was observed in CD8α+ and CD11b+ DCs from cKO^{Kif5b} mice and in Kif5b-deficient BMDCs (Supplementary Fig. 3b). Next, to assess kinesin-1's role in direct presentation, BMDCs from WT or cKO^{Kif5b} mice were infected by the vaccinia virus-encoded OVA epitope and cultured with OT-I T cells. It is noteworthy that the direct presentation of intracellular OVA was not impaired in Kif5b-deficient BMDCs (Supplementary Fig. 4a, b). Moreover, MHC-II presentation (as probed by assaying IL2 production by OT-II T cells) was not affected in DCs lacking Kif5b (Supplementary Fig. 5a). As invariant chain (Ii or CD74) is known to play a key role in MHC-II trafficking and peptide loading, we studied its degradation. WT and cKO^{Kif5b} BMDCs were analysed by western blotting for the presence of Ii and its intermediate degradation products. Ii degradation was not altered in Kif5b-deficient BMDCs (Supplementary Fig. 5b). To gain understanding of the class II-restricted response in cKO^{Kif5b} mice, violet-labelled transgenic OT-II T cells were injected into WT and cKO^{Kif5b}

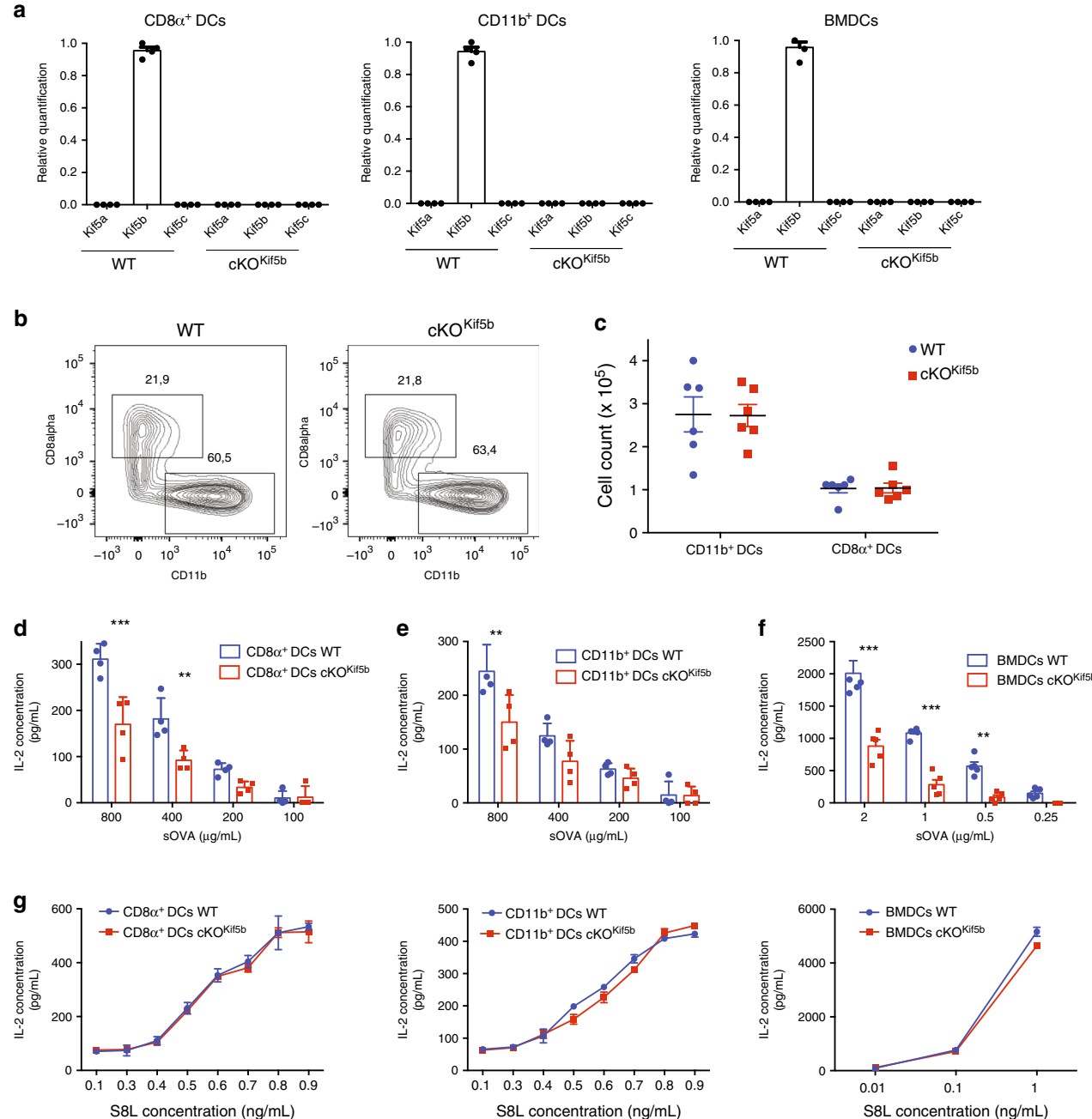

**Fig. 1 Kinesin-1 regulates cross-presentation. a** Relative quantification of Kif5a, Kif5b and Kif5c transcripts by real-time PCR in CD8α$^+$ and CD11b$^+$ DCs and BMDCs from WT or cKO$^{Kif5b}$ mice. Transcript levels for each sample are expressed relative to Kif5b. Graph shows mean ± S.E.M. ($n = 3$). **b** Contour plots of DCs from the spleen of either WT or cKO$^{Kif5b}$ mice pregated on CD19$^-$, Gr$^-$, F4/80$^-$ CD11c$^+$, and IA/IE$^+$. The data are representative of three independent experiments. **c** Absolute numbers of CD11b$^+$ DCs and CD8α$^+$ DCs in the spleen of WT (blue circles) and cKO$^{Kif5b}$ (red squares) mice. The data are representative of five independent experiments. **d–g** The efficiency of cross-presentation of different concentration of sOVA and the presentation of OVA peptide SIINFEKL in vitro by CD8α$^+$ DCs (**d, g**), CD11b$^+$ DCs (**e, g**) and BMDCs (**f, g**) from WT (blue histogram or line) or cKO$^{Kif5b}$ (red histogram or line) mice was measured as IL-2 secretion by OT-I T cells after 16 h of co-culture. Graph shows mean ± S.E.M. ($n = 4$). Statistical significance was determined by the two-way ANOVA and Sidak test's correction for multiple comparison. *$P < 0.05$; **$P < 0.005$; ***$P < 0.0001$.

mice, that were primed the next day with fusion protein containing OVA (P3UOVA), complexed with hamster anti-mouse CD11c antibody (CD11c/P3UOVA) to target the Ag specifically to CD11c$^+$ DCs[23].Three days after priming, the response of OT-II T cells in the spleen was analysed by flow cytometry (Supplementary Fig. 5c). OT-II T cells were equally stimulated in WT and cKO$^{Kif5b}$ mice primed with DC-targeted OVA. Taken as a whole, these results suggest that the defect in T cell priming observed in the absence of kinesin-1 was specific to cross-presentation.

In order to assess the role of cross-presentation in the induction of an adaptive immune response against tumours in vivo, we induced an anti-tumour response in WT and cKO$^{Kif5b}$ mice. Both control and Kif5b-deficient mice were implanted subcutaneously with a highly immunogenic OVA-expressing B16 melanoma cell line. One day later, the mice were injected or not, with OT-I CD8+ T cells. As described previously[24], a significant delay in tumour growth was observed in WT mice injected with OT-I - suggesting that effective OT-I CD8+ T cell priming is

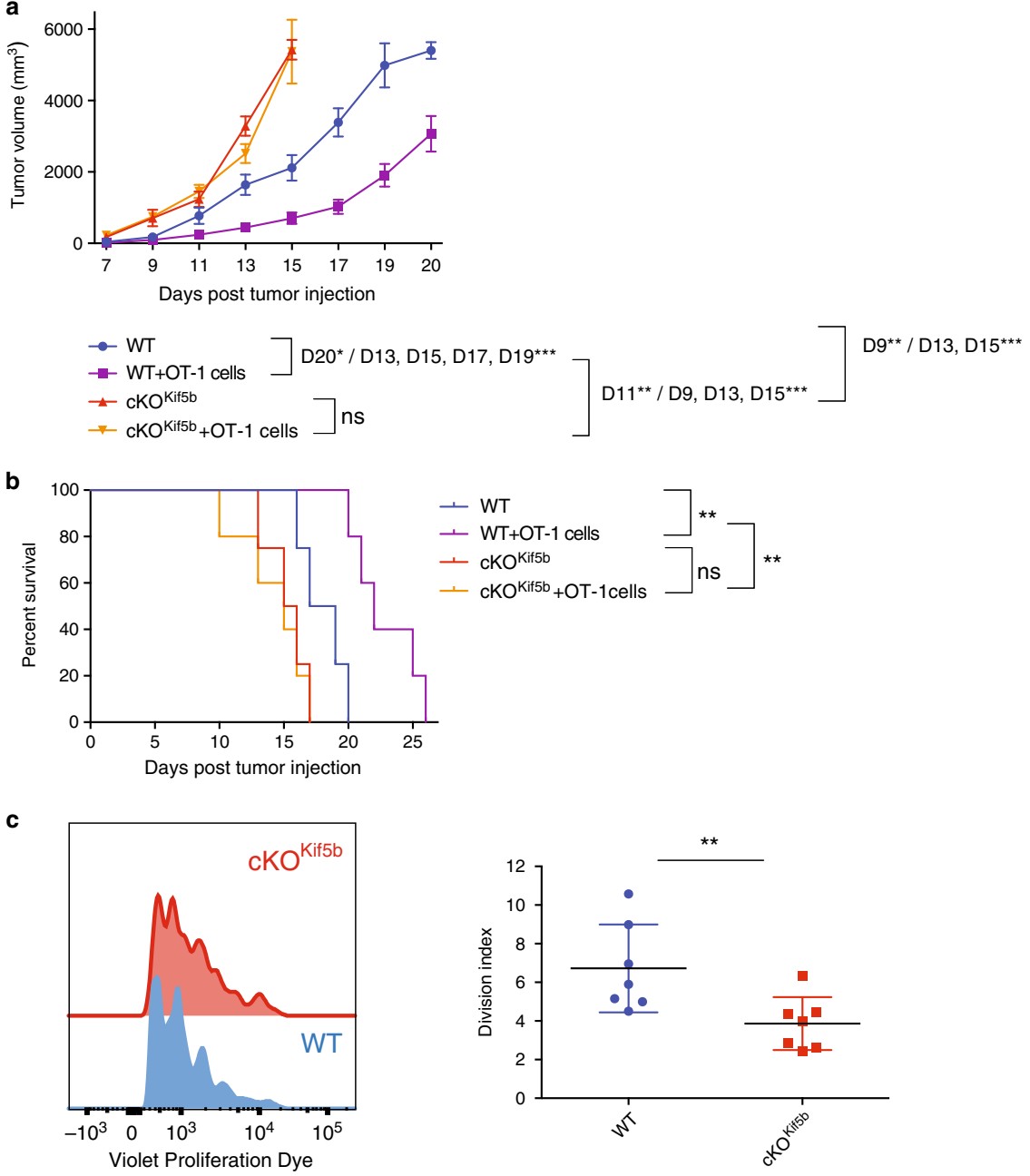

**Fig. 2 cKO^kif5b mice exhibit an impaired tumour response. a** Tumour growth curves for WT (blue line) or cKO^Kif5b (orange line) mice injected subcutaneously with B16-OVA cells and adoptively transferred with OT-I T cells (WT purple line, cKO^Kif5b yellow line). Tumour growth was monitored daily, and non-survival was defined as ulceration of the tumour or a mean tumour diameter of 15 mm. The data are quoted as the mean ± SEM from two independent experiments (WT, WT OT1, cKO^Kif5b $n = 10$ mice per group, cKO^Kif5b OT1 $n = 9$ mice). Statistical significance was determined by the two-way ANOVA and Sidak test's correction for multiple comparison. *$P < 0.05$; **$P < 0.005$; ***$P < 0.0001$. **b** Survival curves for WT or cKO^Kif5b mice, treated as in **a**. Statistical significance was determined by the log-rank test and by the Gehan–Breslow–Wilcoxon test. *$P < 0.05$; **$P < 0.005$. **c** Mice were injected with Violet-labelled transgenic OT-I T cells and primed 1 day later with CD11c/P3UOVA (WT blue line or circles, cKO^Kif5b red line or circles; WT mice $n = 7$, cKO^Kif5b mice $n = 7$). The left panel shows representative profiles gated on CD8^+, TCR Vα2^+ cells. Statistical analysis of the division index is shown in the right panel: **$P < 0.005$ in a two-tailed unpaired Student's $t$ test. Graph shows mean ± S.E.M.

mediated by DCs through cross-presentation (Fig. 2a). Interestingly, B16-OVA tumours grew faster in cKO^Kif5b mice (Fig. 2a) and reduced survival (Fig. 2b), when compared with WT mice. Ten days following tumour injection, no difference in the number of CD8+ T cells, CD4+ T cells, B cells and NK cells was observed in cKO^Kif5b and WT mice (Supplementary Fig. 6a). The NK cell cytotoxic function was not altered in the absence of Kif5b as NK WT vs. NK cKO^Kif5b did not differ in their ability to degranulate

cytotoxic granules (Supplementary Fig. 6b). Strikingly, the adoptive transfer of naïve OT-I T cells to cKO^Kif5b mice did not prevent tumour progression and did not improve survival (Fig. 2a, b) – suggesting that a defective anti-tumour response in cKO^Kif5b mice could be due to inefficient Ag cross-presentation. To confirm the inefficient Ag cross-presentation in vivo in cKO^Kif5b mice, violet-labelled transgenic OT-I T cells were injected into WT and cKO^Kif5b mice that were primed the next

day with CD11c/P3UOVA[23]. Three days after priming, the response of OT-I T cells in the spleen of WT and cKO[Kif5b] mice was analysed by flow cytometry (Fig. 2c). These data show a lower rate of proliferation of OT-I T cells in the spleen of cKO[Kif5b] mice compared to WT mice, confirming the compromised ability of Kif5b-deficient DCs to cross-prime in vivo.

We conclude that kinesin-1 deficiency in both splenic DCs and BMDCs impairs cross-presentation in vitro and in vivo. In contrast, the presentation of endogenous Ag to MHC-I molecules and the presentation of exogenous Ag to MHC-II molecules in cKO[Kif5b] mice was unaffected.

**Impaired Ag degradation and acidification in cKO[Kif5b] endosomes.** To gain insights into kinesin-1's role in cross-presentation, we first tested its involvement in Ag uptake. CD8α+ and CD11b+ DCs purified from the spleen of cKO[Kif5b] mice and Kif5b-deficient BMDCs did not differ in their ability to internalise sOVA or to phagocytise OVA-beads, relative to control CD8α+ and CD11b+ DCs and BMDCs, respectively (Fig. 3a, Supplementary Fig. 7). Hence, kinesin-1 does not appear to control Ag internalisation (Fig. 3a, Supplementary Fig. 7). We next looked at whether the decrease in cross-presentation ability might be due to slow Ag degradation. Flow cytometry was used to monitor the degradation of sOVA labelled with a pH-insensitive boron-dipyrromethene fluorescent dye (DQ-OVA) that exhibits bright green fluorescence upon proteolytic processing. After the cells had been pulsed with DQ-OVA, Ag degradation was assessed at different time points. Antigen degradation was weaker in purified spleen CD8α+ and CD11b+ DCs from cKO[Kif5b] mice and Kif5b-deficient BMDCs than in WT CD8α+ and CD11b+ DCs and WT BMDCs (Fig. 3b). The defect in Ag degradation was also observed for a particulate Ag (Supplementary Fig. 8). To assess kinesin-1's role in Ag degradation, we used specific inhibitors of endosomal protease activities (a cathepsin inhibitor which inhibits cathepsin B, cathepsin L, cathepsin S and papain) and endosomal pH acidification (concanamycin A, a V-ATPase inhibitor). Incubation with either of the inhibitors inhibited Ag degradation in WT cells but essentially had no effect in the Kif5b-deficient BMDCs (Fig. 3c). Furthermore, both inhibitors decreased Ag degradation in WT BMDCs to the level observed in Kif5b-deficient BMDCs, underlining the dramatic impairment of endocytic protein degradation in Kif5b-deficient cells.

To determine whether or not the Ag degradation defect in Kif5b-deficient DCs reflected a change in endosomal acidification, we measured the endosomal pH at different time points. As shown in Fig. 3d, the endosomal pH was more alkaline (by ~1 pH unit) in CD11b+ DCs from cKO[Kif5b] mice and Kif5b-deficient BMDCs than it was in WT CD11b+ DCs and BMDCs. However, no pH difference was observed when comparing CD8α+ DCs from cKO[kif5b] mice with WT DCs; this lack of a difference might be due to an intrinsically higher endosomal pH in CD8α+ DCs than in CD11b+ DCs[6].

To investigate whether the observed differences in the Ag degradation and the endosomal pH between WT BMDCs and Kif5b-deficient BMDCs may be linked to the variations in the endosomal proteolytic activity, we assessed the activity of cathepsins B and L (catB/L) in total cell lysates. As shown in Fig. 3e, the proteolytic activity of catB/L in the total lysate of Kif5b-deficient BMDCs was markedly decreased at 20 and 120 min. These results indicate that the change in the endosomal pH in Kif5b-deficient DCs can modify the activity of endosomal proteases and suggest a defect in Ag degradation as a result.

**Kinesin-1 is required for sOVA trafficking and MHC-I recycling.** In order to analyse kinesin-1's role in the vacuolar Ag

degradation process, sOVA trafficking along the endocytic pathway was examined by immunofluorescence microscopy. We first confirmed that in the absence of Kif5b, the number and morphology of the different endosomal compartments, Golgi apparatus and endoplasmic reticulum were unaltered (Supplementary Fig. 9). Next, WT and cKO[Kif5b] BMDCs were pulsed with Alexa Fluor 647-labelled sOVA for 15 min and chased for different time points (15, 30 or 60 min). Wild-type and cKO[Kif5b] BMDCs were then stained with different endosomal markers, such as EEA1 for early endosomes, Lamp1 for late endosomes/lysosomes, and Rab11 for recycling endosomes. In the absence of kinesin-1, we observed a delay in endosomal maturation: in Kif5b-deficient BMDCs, the percentage of colocalization of sOVA with EEA1 was higher at 15 min and 60 min and the percentage of colocalization sOVA with LAMP1 was lower at 15 min and higher at 60 min than in WT BMDCs, the latter suggesting a delay in OVA recruitment in Lamp1+ compartment in Kif5b-deficient BMDCs (Fig. 4). Furthermore, very little sOVA reached the recycling compartment during the 60 min kinetic study in Kif5b-deficient BMDCs (Fig. 4). These results indicate that kinesin-1 regulates the localisation of sOVA in endosomal compartments by facilitating the Ag's recruitment to recycling and late endosomal compartments.

Although poorly characterised in murine DCs, MHC-I recycling is thought to be important in cross-presentation, by providing MHC-I molecules that are loaded with cross-presented peptides. To address this, we adopted a published assay[25] that is designed to measure the rate of recycling indirectly by quantifying the percentage of total cellular MHC-I lost upon acid stripping of cell surface molecules. We first assessed MHC-I cell surface expression in CD8α+ and CD11b+ DCs and BMDCS from WT or cKO[Kif5b] mice. There were no apparent differences in cell surface MHC-I expression between WT and cKO[Kif5b] DCs and BMDCs (Fig. 5a). Using flow cytometry, we next evaluated the presumable MHC-I recycling capacities of CD8α+ and CD11b+ DCs and BMDCs from WT or cKO[Kif5b] mice. We observed a significant reduction in the loss of total cellular MHC-I after acid stripping in Kif5b-deficient DCs, relative to WT DCs (Fig. 5b). This defect was not restricted to MHC-I because a similar impairment in recycling was evidenced for the transferrin receptor (CD71) (Fig. 5c). To confirm the recycling defect in the absence of Kif5b, we performed a second recycling assay including a primaquine treatment, that was reported to slow the return of receptors on the cell surface by increasing their intracellular pool[26]. MHC-I reappearance at the cell surface, after 30 min of internalisation in the presence of primaquine followed by another incubation for up to 30 min to allow the export of accumulated MHC-I, was drastically impaired in Kif5b-deficient BMDCs compared to WT BMDCs (Fig. 5d). Collectively, these results show that kinesin-1 is required for MHC-I recycling.

We next analysed the colocalization of MHC-I with endocytic markers in BMDCs generated from WT or cKO[Kif5b] mice, using immunofluorescence. In the absence of Kif5b, the percentage of colocalization of MHC-I with EEA-1 was higher and the percentage of colocalization of MHC-I with Rab11 was lower, when compared with WT BMDCs (Fig. 5e). The latter result might indicate that kinesin-1 is required for MHC-I recruitment from early endosomes to recycling endosomes.

**Kinesin-1 regulates early endosome dynamics.** In order to determine how kinesin-1 regulates endosomal vesicular trafficking, we first used confocal microscopy to investigate Kif5b's localisation in BMDCs. Endogenous Kif5b was mainly distributed along the microtubules and around the microtubule-organising centre (Kif5b/tubulin overlap: 76.18% ± 5.35%, Fig. 6A). Kif5b-deficient

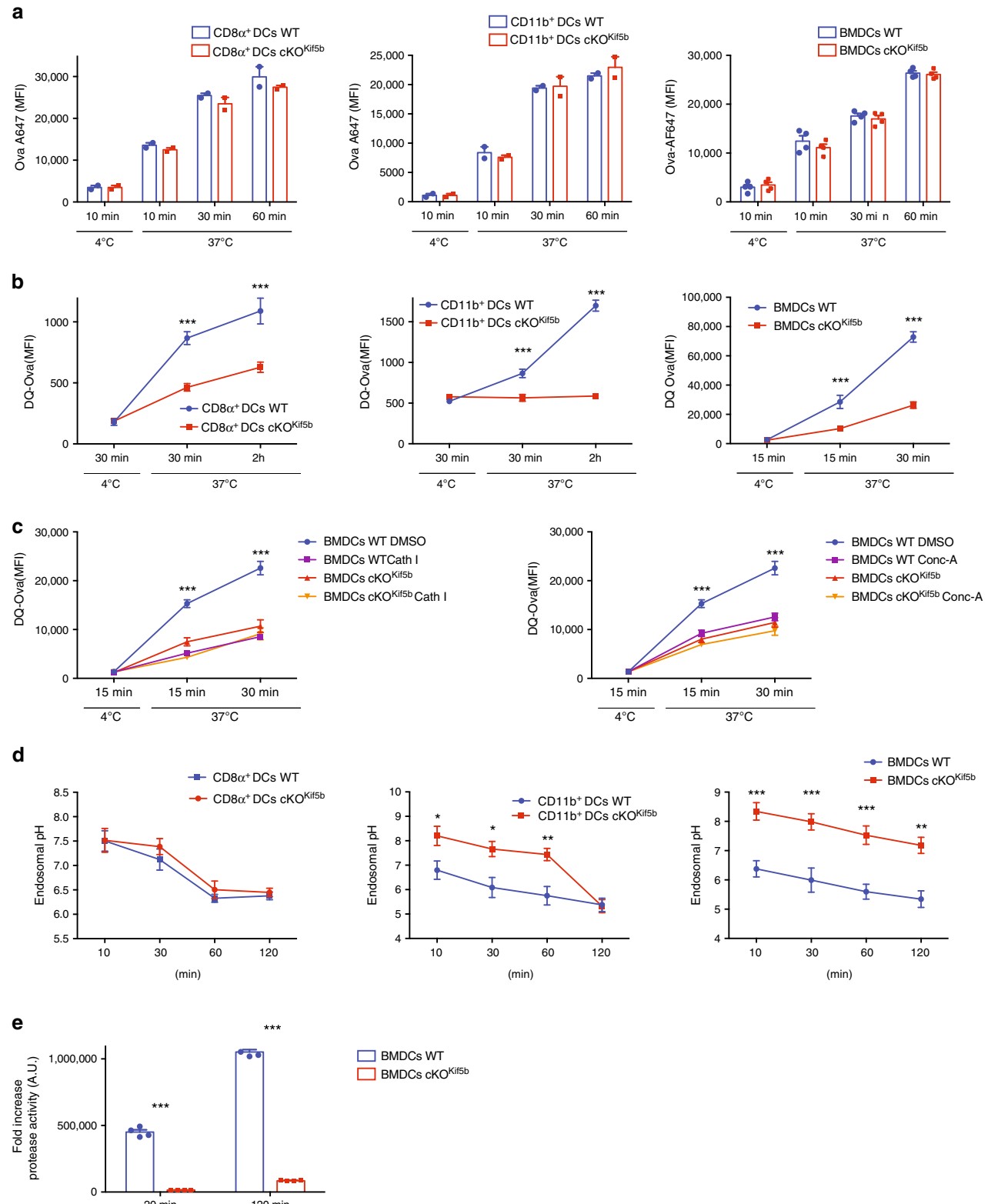

BMDCs did not display any Kif5b labelling - confirming the antibody's specificity (Fig. 6a). Confocal immunofluorescence images revealed greater colocalization of Kif5b with early endosomes than with recycling endosomes and late endosomes (EEA1/Kif5b overlap: 53.7% ± 7.7% (Fig. 6b); Lamp1/Kif5b overlap: 36.1% ± 6.7% (Supplementary Fig. 8); Rab11/Kif5b overlap: 29.8% ± 7.5% (Supplementary Fig. 10)); these results suggested that kinesin-1 can regulate early endosomal compartment trafficking. To assess

kinesin-1's role in the dynamics of early compartment motility, we used live cell spinning disk microscopy. After a brief period of wheat germ agglutinin (WGA) endocytosis (pulsed for 5 min and chased for 5 min), WGA was associated with the early endosomes and colocalized with EEA1-positive early endosomes in BMDCs from both WT or cKO^Kif5b mice (EEA1/WGA overlap: 81.7% ± 5.7% in WT BMDCs, and 83.7% ± 5.7% overlap in cKO^kif5b BMDCs; Fig. 6c, d). Upon WGA loading, spinning disk microscopy

**Fig. 3 Kinesin-1 regulates Ag degradation and acidification in the endocytic pathway. a** CD8α[+], CD11b[+] DCs and BMDCs from WT (blue histogram) or cKO[Kif5b] (red histogram) mice were incubated for 10, 30 or 60 min at 37 °C or for 10 min at 4 °C with Alexa-Flour-647-OVA prior to flow cytometry analysis. The figure shows a histogram that was representative of three independent experiments. **b** CD8α[+], CD11b[+] DCs and BMDCs from WT (blue line) or cKO[Kif5b] (red line) mice were incubated for 30 min at 37 °C with DQ-OVA. Next, the cells were washed. DQ-OVA degradation was analysed by flow cytometry after different chase periods. Graphs are representative of four independent experiments. Statistical significance was determined by the two-way ANOVA and Sidak test's correction for multiple comparison. ***$P < 0.0001$. **c** Degradation of DQ-OVA in BMDCs from WT (DMSO blue line, Conc-A or Cath I purple line) or cKO[Kif5b] (DMSO orange line, Conc-A or Cath I yellow line) mice pretreated or not with concanamycin A (Conc-A) or a cathepsin inhibitor-I (Cath I). Graphs are representative of four independent experiments. Statistical significance was determined by the two-way ANOVA and Sidak test's correction for multiple comparison. **$P < 0.005$; ***$P < 0.0001$. **d** The pH values in CD8α[+], CD11b[+] DCs and BMDCs from WT (blue line) or cKO[Kif5b] (red line) mice were determined by FACS after a pulse and different chase periods. Graphs are representative of four independent experiments. Statistical significance was determined by the two-way ANOVA and Sidak test's correction for multiple comparison. *$P < 0.05$; **$P < 0.005$; ***$P < 0.0001$. **e** Protease activity of CatB/L in early (20 min) or late (120 min) using total cell lysate from BMDCs from WT (blue histogram) or cKO[Kif5b] (red histogram) mice was measured with a specific fluorescent substrate. Graphs are representative of three independent experiments. Statistical significance was determined by the two-way ANOVA and Sidak test's correction for multiple comparison. ***$P < 0.0001$. **a–e** Graphs show mean ± S.E.M.

images were acquired for 5 min. In WT BMDCs, the WGA-labelled early endosomes were very dynamic, and displayed by several characteristic vesicle fusion and fission events (Fig. 6e and Supplementary Movie 1). In contrast, the early endosomes were less dynamic in Kif5b-deficient BMDCs. We observed the formation of tubulations that were subsequently unable to detach from the endosomal membrane (Fig. 6e, f and Supplementary Movie 2). To further characterise the early endosomes' dynamics, we monitored the size of the 1-μm vesicular structure during the 5-min acquisition period. The early endosomes grew more quickly in Kif5b-deficient BMDCs than in WT BMDCs, which confirmed the imbalance between the fusion and fission of early endosome compartments in the absence of Kif5b (Fig. 6g). To monitor the time course of early endosome recruitment to late endosomes/lysosomes in BMDCs from WT or cKO[Kif5b] mice, the cells were incubated with Alexa-Flour-555-WGA for a longer period in order to label late endosomes/lysosomes in red (WGA-555 pulsed for 5 min and then chased for 2 h). After the chase, the Alexa-Flour-555-WGA was found to be associated with the late endosomes/lysosomes (i.e. colocalized with Lamp1) in BMDCs from WT or cKO[Kif5b] mice (Lamp1/WGA overlap: 81.7% ± 7.4% in WT BMDCs and 85.11% ± 4.4% in cKO[Kif5b] BMDCs) (Supplementary Fig. 11). Next, the same cells were pulsed with Alexa-Flour-488-WGA to label early endosomes in green (Fig. 6h). Recruitment of early endosomes to late endosomes/lysosomes was monitored by acquiring live cell spinning disk microscopy images every 5 min for 120 min. The recruitment appeared to be slower in Kif5b-deficient BMDCs than in WT BMDCs (Fig. 6h, i). Taken as a whole, these data indicate that Kif5b can be recruited on EEA1-positive sorting endosomes, drives the scission of tubular structures, enables vesicles to mature into recycling or late endosomes, and thus may participate to the subsequent sorting of Ag/MHC-I complexes.

**Microtubules are required for tubulation scission.** Given that kinesin-1 is a microtubule-based motor, we assessed the role of microtubules in endosomal tubule extension and fission. We first looked at whether the microtubule network is required for Ag internalisation and degradation by performing experiments with nocodazole, an agent that promotes tubulin depolymerisation. Disruption of the microtubule network by nocodazole was confirmed by tubulin staining (Supplementary Fig. 12A). Control (DMSO-treated) BMDCs and nocodazole-pre-treated BMDCs internalised sOVA to the same extent (Fig. 7a). In contrast, sOVA were less extensively degraded by nocodazole-pre-treated BMDCs than by DMSO-pre-treated BMDCs (Fig. 7b). Given that nocodazole impaired Ag degradation, we next assessed the microtubules' role in early endosome dynamics by using labelled WGA and live cell spinning disk microscopy. WGA was found to colocalize with EEA1-positive early endosomes in both nocodazole-treated BMDCs

and DMSO-treated BMDCs (Supplementary Fig. 12b). However, nocodazole-treated BMDCs exhibited extensive tubulations that were unable to detach from the endosomal membrane - mimicking the observations of Kif5b-deficient BMDCs (Fig. 7c, d and Supplementary Movies 3 and 4). These data suggest that the microtubule network is required for tubulation fission but not for tubulation formation.

Since actin polymerisation is involved in tubule formation and the stabilisation of endosomal membranes[27], we next evaluated its role in the formation of endosomal tubulations. We first assessed actin's role in Ag uptake and degradation. Incubation of BMDCs with the F-actin depolymerising agent latrunculin B (LatB) depleted actin fibres (stained with phalloidin-A647; Supplementary Fig. 12a) and impaired Ag degradation, whereas Ag internalisation was not affected (Fig. 7a, b). In LatB-treated BMDCs, WGA-labelled EEA1-positive early endosomes were found to be enlarged (Supplementary Fig. 12b, c). Video microscopy analysis of LatB-treated BMDCs revealed an absence of tubule formation, which prevented the formation of individual vesicles (Fig. 7c, e and Supplementary Movie 5). These data show that actin polymerisation is required for the formation and elongation of the tubulation, whereas microtubules and kinesin-1 are required for tubule detachment and thus vesicle maturation.

## Discussion

Antigen cross-presentation by DCs is known to have a crucial role in the initiation of adaptive cytotoxic immune responses against pathogens and tumours. Although cross-presentation depends critically on trafficking of intracellular vesicular compartments, the molecular machinery that mediates the transport of Ag-containing vesicles is poorly understood. The present study provided in vivo and in vitro evidence to show that the Kif5b heavy-chain isoform of kinesin-1 is involved in Ag cross-presentation by DCs.

Indeed, a lack of Kif5b dramatically impaired (i) IL-2 secretion from OT-I T cells upon cross-presentation of sOVA or particulate OVA by DCs, (ii) an in vivo anti-tumour response, (iii) the degradation of soluble or particular Ags, (iv) acidification and proteolysis activity of the endocytic pathway, (v) sOVA intracellular trafficking, (vi) MHC-I recycling and (vii) early endosome dynamics, including the scission of tubulations from early endosomes (Supplementary Fig. 13). In contrast, MHC-II presentation, endogenous MHC-I Ag presentation through the classical secretory pathway, and Ag internalisation were found to be independent of Kif5b.

In previous studies, we showed that kinesin-1 is required for the secretion of specialised secretory granules in cytotoxic T cells, mast cells and platelets[17–19], corresponding to kinesin-1-based transport of late endosomes/lysosomes. A role for Kif5b in the

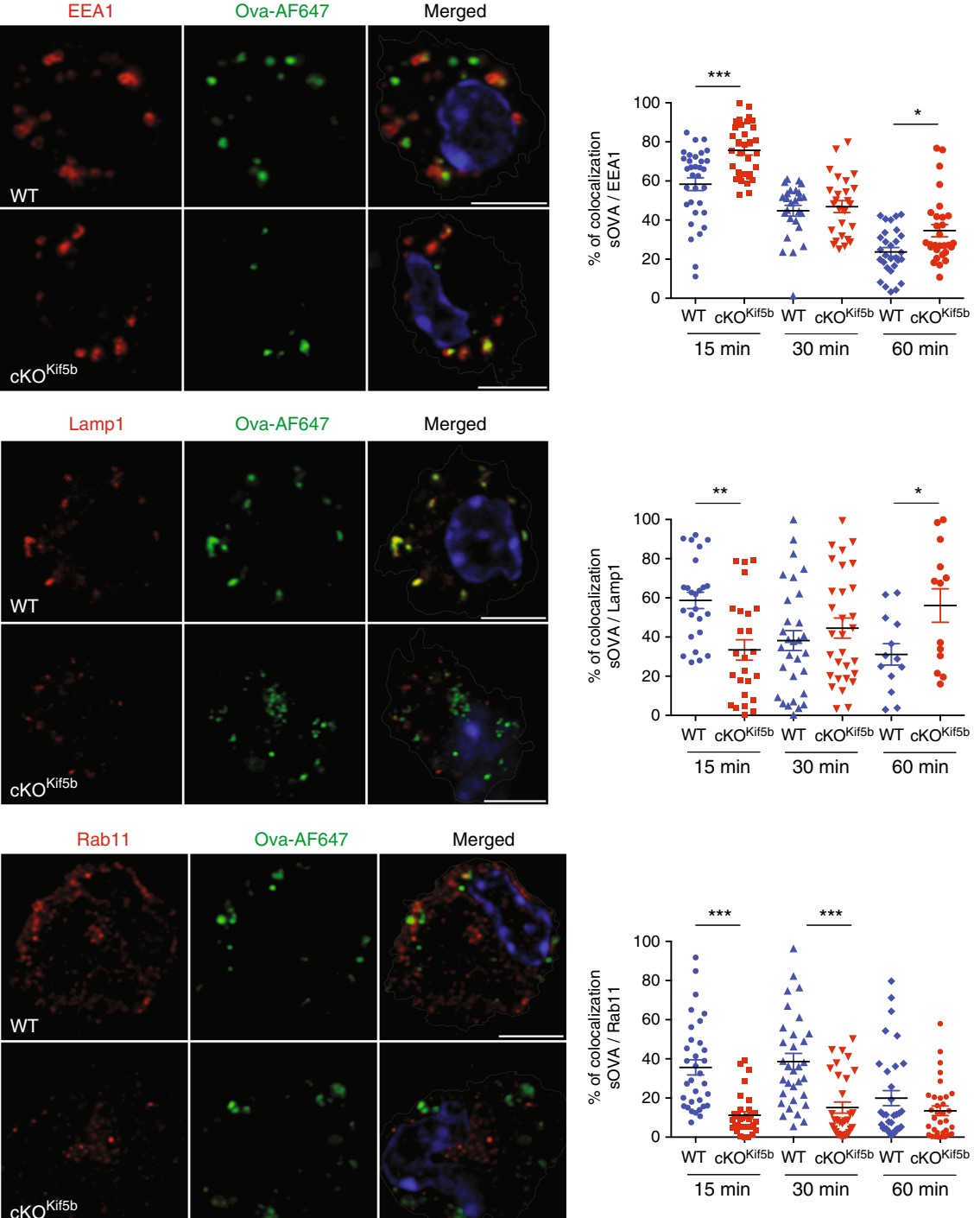

**Fig. 4 Kinesin-1 regulates the intracellular trafficking of sOVA.** BMDCs from WT or cKO$^{Kif5b}$ mice were incubated for 15, 30 or 60 min with Alexa-Flour-647-OVA and then plated on glass coverslips. The cells were fixed, permeabilized and stained with anti-EEA1, anti-Lamp1 or anti-Rab11 antibodies. The left panel shown representative images for the 15 min time point observed in three independent experiments. Bars: 5 μm. $n = 32$ cells per condition. Statistical significance (shown in the right panel; WT blue circle, cKO$^{Kif5b}$ red square) was determined by the two-way ANOVA and Sidak test's correction for multiple comparison. **$P < 0.005$; ***$P < 0.0001$. Graphs show mean ± S.E.M.

distribution of early endosomes and in endosome fission has been previously suggested in a couple of studies[21,22]. Our data show that in DCs, kinesin-1 regulates early endosomal compartments by controlling vesicular trafficking dynamics and allowing cargo sorting for recycling and/or degradation. Indeed, the cross-presentation defect observed here in the absence of Kif5b correlated with an impairment in early endosome dynamics and membrane fission. Moreover, we demonstrated that in Kif5b-

deficient DCs, the extensive tubulation structures were unable to detach from the early endosomal compartments. This finding suggests that kinesin-1 is not the microtubule motor required for extensive tubule formation along the microtubule track but is absolutely essential (by recruiting additional effectors and/or by pulling tubules) to induce tubule detachment for intracellular vesicular transport. It was recently reported that the dynamin-related GTPase Mx1 controls endosomal elongation and fission in

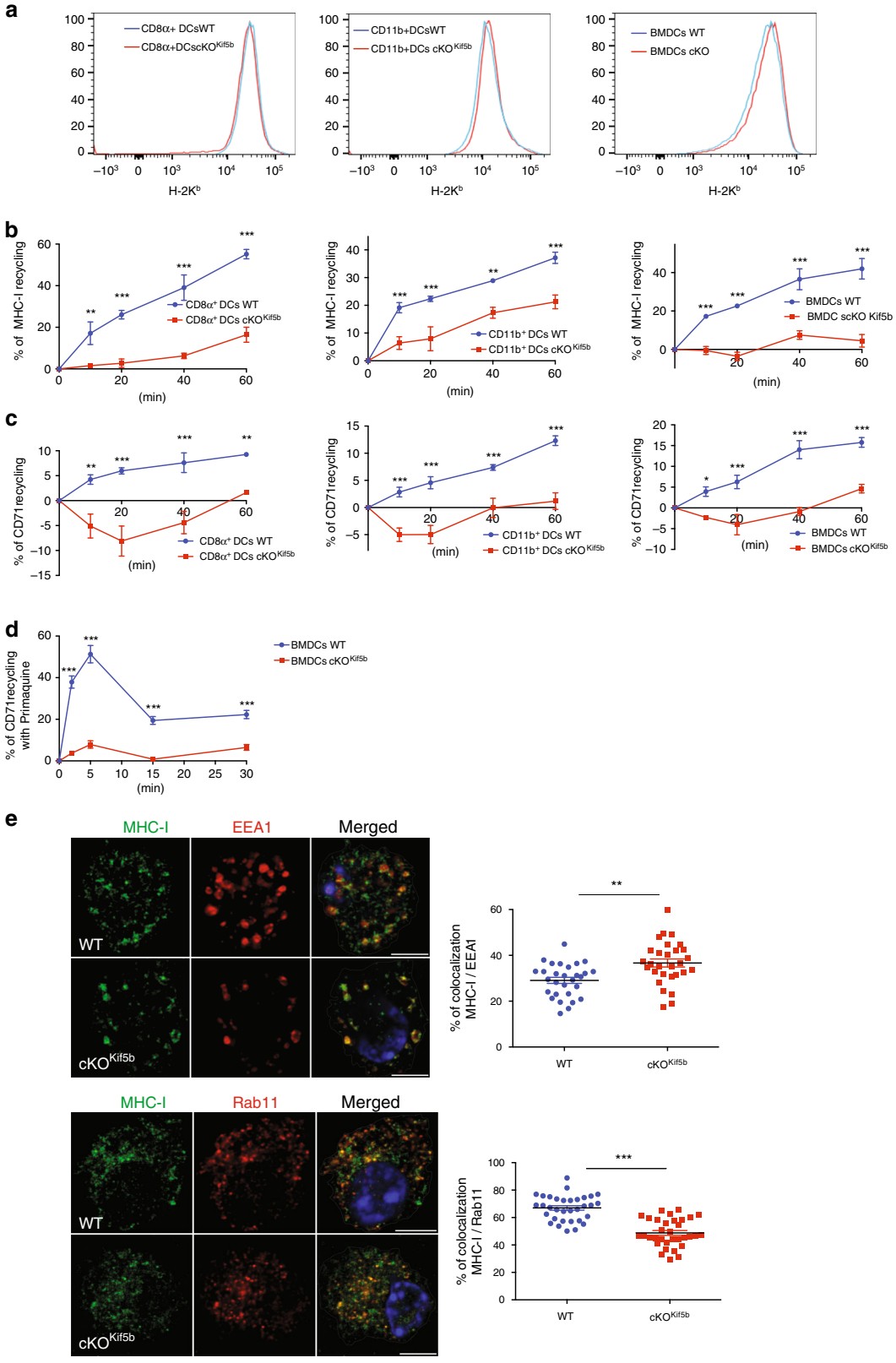

epithelial cells by recruiting Kif5b and thus enabling the apical transport of vesicles[28]. Thus, future work should assess whether Mx1 or an equivalent recruiting pathway could be involved in Kif5b recruitment on early endosomes in DCs.

Interestingly, it has been hypothesised that tubular formation and fission may be induced by endosome fusion, since both processes can maintain compartment homeostasis[29]. In agreement with this hypothesis, the absence of tubular fission is associated with a rapid increase in the size of early endosomes. These observations illustrate the importance of regulating the fusion and fission processes.

Even though kinesin-1 was not found to be involved in membrane tubule extension in DCs, the kinesin superfamily is

**Fig. 5 Kinesin-1 regulates MHC-I recycling. a** CD8α[+], CD11b[+] DCs and BMDCs from WT (blue line) or cKO[Kif5b] (red line) mice were labelled for H-2K[b] and analysed using FACS. The figure shows a FACS histogram that was representative of three independent experiments. **b** MHC-I recycling ability in CD8α[+], CD11b[+] DCs and BMDCs from WT (blue line) or cKO[Kif5b] (red line) mice was measured using FACS at the indicated time points. Statistical significance was determined by the two-way ANOVA and Sidak test's correction for multiple comparison. **P < 0.005; ***P < 0.0001. **c** TrfR (CD71) recycling ability was measured using FACS at the indicated time periods in CD8α[+], CD11b[+] DCs and BMDCs from WT or cKO[kif5b] mice. Statistical analysis: *P < 0.05; **P < 0.005; ***P < 0.0001 in the two-way ANOVA and Sidak test's correction for multiple comparison. **d** MHC-I recycling in the presence of primaquine in BMDCs from WT or cKO[Kif5b] mice was measured using FACS at the indicated time points. Statistical analysis: **P < 0.005; in the two-way ANOVA and Sidak test's correction for multiple comparison. **b–d** Graphs are representative of four independent experiments. **e** BMDCs from WT or cKO[kif5b] mice were plated on glass coverslips and were fixed, permeabilized and stained with anti-EEA1, anti-Rab11 or anti-MHC-I antibodies. Representative images are shown in the left panel observed in three independent experiments. (n = 33 cells per condition upper panel, n = 29 cells per condition lower panel) Bars: 5 μm. Statistical analysis (shown in the right panel; WT blue circle, cKO[Kif5b] red square): **P < 0.005; ***P < 0.0001 in an two-tailed unpaired Student's t test. **b–e** Graphs show mean ± S.E.M.

known to provide the driving force for the membrane tubulation and fission required for cargo trafficking in some cell types[30]. Some of the kinesin-mediated membrane extension functions have been shown to involve members of kinesin-3 family. Indeed, the endosome-binding microtubule motor Kif16B belongs to the kinesin-3 family and has been identified as a candidate for pulling a membrane tubulation. Kif16B reportedly mediates the movement of early endosomes - an event that was found to be required for Tfn recycling and transport from early to late endosomes[31]. Interestingly, it was recently reported that Kif16B is involved in cross-presentation via the recruitment of Rab14 upon innate immune signalling through Toll-like receptor 4 or through Fc receptors[32]. Kif13A (another member of the plus-end-directed kinesin-3 family) has been shown to be essential for generating endosomal tubules in a microtubule- and motor-dependent manner by pulling Rab11-positive endosomal tubules[33].

We therefore sought to determine whether the generation of the driving force for the extensive tubulation seen in early endosomes required the actin cytoskeleton or the microtubule network. We found that actin was essential for the generation and then elongation of tubulations, whereas microtubules and kinesin-1 were required for tubule detachment (Supplementary Fig. 13). It would now be useful to determine which actin-based motors generate the driving force of the extensive tubulation of early endosomes in DCs.

The current model of vesicle scission predicts that the Bar (Bin/amphiphysin/Rvs) protein scaffold, microtubule-based molecular motors and actin polymerisation generate the force required for membrane fission[34]. In melanocytes, the detachment of tubules from recycling endosomes enables cargo recycling and the biogenesis of pigment granules[27,35]. Constriction and fission of tubules in melanocytes is mediated by BLOC1, annexin A2, KIF13A, Myo6 and WASH- and ARP2/3-dependent actin polymerisation that act on microtubule- and actin-dependent machineries[27,35].

Given that kinesin-1 has been shown to regulate the transport of Rab27/Slp-associated secretory granules on microtubules[17–19], it is possible that Rab27a might also regulate the early endosome dynamic in DCs. Interestingly, Rab27a deficiency has been linked to excessive particular Ag degradation via the impairment of NOX2 recruitment to phagosomes[36]. Considering that Rab27a- and Kif5b-deficiencies have opposite effects on Ag degradation, it is tempting to speculate that Rab27a does not participate in the fission of early endosomes. Future research should seek to determine which Rab GTPases are able to recruit kinesin-1 and/or whether other Kif5b effectors are responsible for kinesin-1's recruitment to EEA1-positive endosomal tubules.

Another intriguing finding was that kinesin-1 is required for intracellular trafficking of sOVA and MHC-I molecules. We propose that a primary defect in early endosomal membrane

fission results in delayed maturation of early endosomes, that impairs delivery of internalised antigens to degradative compartments and the recruitment of MHC-I from early endosomes to recycling endosomes. Consequently, kinesin-1 is required for sorting recently internalised cargoes (including Ags, MHC-I molecules and receptors) towards the recycling endosomal or lysosomal compartments. However, it is not clear which of these defects are responsible for impaired Ag cross-presentation. Several hypotheses can be raised. First, the drastic defect in Ag degradation, that impairs the generation of proteolytic peptides derived from Ag, could account alone for the defective Ag cross-presentation. Even though, it is now well established that increased Ag degradation has a detrimental effect on efficient cross-presentation[5,36–38], it is possible that a profound impairment of the Ag degradation will also impact cross-presentation. Indeed, this is illustrated by the effect of removing the cysteine protease cathepsin S that significantly affects cross-presentation of some Ag forms in vitro and in vivo[12]. Alternatively, the Ag cross-presentation defect could be attributed mainly to defective endosomal and MHC-I trafficking and recycling. The lack of MHC-II Ag presentation defect supports this hypothesis as the Ag degradation observed in Kif5b-deficient DCs has no detectable effect on the MHC-II Ag presentation pathway. A third hypothesis is an additive effect of defective Ag degradation and MHC-I recycling in Ag cross-presentation impairment.

We were intrigued by the finding that the MHC-II Ag presentation was not impacted by the absence of Kif5b. Apparently, reduced and delayed Ag degradation defect evidenced in Kif5b-deficient DCs does not perturb with the same severity as the MHC-II pathway. We cannot rule out the possibility that other kinesins regulate MHC-II trafficking or compensate for the loss of kinesin-1. As discussed above, we cannot exclude either, that the defect in Ag degradation will not be responsible for defective Ag cross-presentation and similarly will not affect MHC-II Ag presentation.

In conclusion, we used a murine conditional knockout model (lacking Kif5b in all hematopoietic lineages) to demonstrate that kinesin-1 regulates Ag cross-presentation in the cDC1 and cDC2 subsets and thus is essential for an effective CD8+ T-cell-mediated anti-tumour response. In this context, kinesin-1 was found to be essential for Ag degradation, acidification and proteolysis activity of the endocytic pathway, and MHC-I recycling by regulating early endosome dynamics. By controlling Ag and MHC-I sorting via membrane tubulation fission, kinesin-1 appears to be a molecular checkpoint that modulates the balance between Ag degradation and cross-presentation. Kinesin-1 can thus be considered as a potential target for modulating cross-presentation and amplifying immune responses.

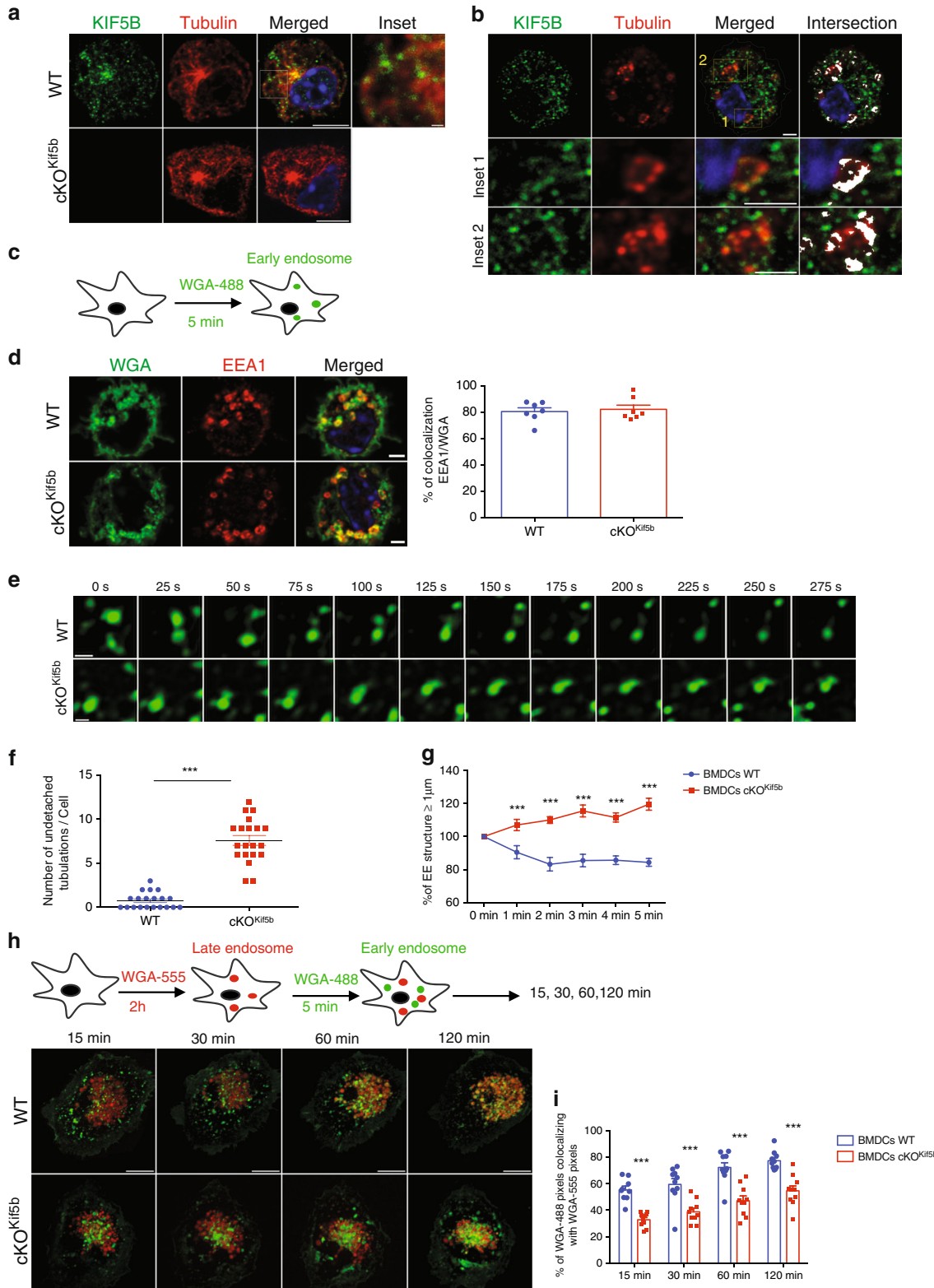

## Methods

**Reagents and antibodies.** Anti-CD11c (BV711 (BioLegend), APC (BD Biosciences) PB (Sony)), anti-IA/IE FITC (BD Biosciences), anti-CD8α BV450 (Sony), anti-CD11b (PE (BD Biosciences), APC-CY7 (Sony)), anti-F4/80 PE/Cy7 (Sony), anti-GR1 APC (BD Biosciences), anti-CD19 (PerCP (BD Biosciences), PerCP/Cy5.5 (Sony)), anti-B220 BV650 (BioLegend), anti-CD86 BV650 (Sony), anti-CD40 PE/Cy7 (Sony), mouse monoclonal Kif5b antibody (Biolegend), purified rabbit anti-α-tubulin (Rockland), goat anti-EEA1 antibody (Santa Cruz Biotechnology), rat anti-Lamp1 antibody (Invitrogen), H-2Kb PE antibody (Biolegend), AF6-88.5 clone (Biolegend), rabbit Rab11 antibody (Life Technologies), anti-CD71 PE antibody (Sony), Alexa Fluor-594 donkey anti-rat IgG (H+L) (Life Technologies), Alexa Fluor-555 donkey anti-goat IgG (H+L) (Life Technologies), Alexa Fluor-488 donkey anti-mouse IgG (H+L) (Life Technologies), and Alexa Fluor-555 donkey anti-rabbit IgG (H+L) (Life Technologies) were used. Alexa Fluor-488 and -555 WGAs were purchased from Invitrogen. Violet Proliferation Dye 450 was purchased from BD Biosciences. Concanamycin was purchased from Sigma, and cathepsin inhibitor-I was purchased from Calbiochem; the inhibitors were used at a concentration of 4 nM and 100 μM, respectively.

**Fig. 6 Kinesin-1 regulates early endosome dynamics. a** Confocal microscopy of BMDCs from WT or cKO$^{Kif5b}$ mice stained with anti-tubulin and anti-Kif5b. Bars: 5 μm. The indicated box is shown at a higher magnification in the inset. Bars: 2 μm. **b** Confocal microscopy of WT BMDCs stained with anti-EEA1 and anti-Kif5b. The intersection between the two stainings is shown in white. The indicated boxes are shown at higher magnification in the insets. Bars: 2 μm. **c** A schematic representation of early endosome WGA labelling. **d** WT and cKO$^{Kif5b}$ BMDCs were labelled with WGA-488 and stained with anti-EEA1. Bars: 2 μm (left panel). Quantification of the colocalization between WGA-488 and EEA1-555 (right panel; WT blue histogram, cKO$^{Kif5b}$ red histogram; $n = 7$ cells per condition). All images of single cells are representative of >100 cells observed in three independent experiments (**a–d**). **e** Spinning disk video microscopy was performed on WT and cKO$^{Kif5b}$ BMDCs labelled with WGA-488. Representative series of images are shown every 25 s. Bars: 2 μm. Images are representative of four independent experiments. See also Supplementary Movies 1 and 2. **f** The number of tubulations unable to detach per cell during the 5 min acquisition. (WT blue circles, cKO$^{Kif5b}$ red squares; $n = 20$ cells per condition). Statistical analysis: ***$P < 0.0001$ in a two-tailed unpaired Student's $t$ test. **g** The number of WGA-488-positive vesicular structures >1 μm in size was measured in WT (blue line) and cKO$^{Kif5b}$ (red line) BMDCs during the 5 min acquisition ($n = 13$ cells per condition). Statistical analysis: ***$P < 0.0001$ in a two-way ANOVA and Sidak test's correction for multiple comparison. **h** Schematic representation of the WGA labelling protocol (upper panel). Spinning disk video microscopy was performed on WT and cKO$^{Kif5b}$ BMDCs labelled with WGA-555 and WGA-488. Representative series of images observed in three independent experiments are shown at the indicated time periods (lower panel). Bars: 2 μm. **i** $n = 10$ cells per condition. Statistical analysis of the experiment in (WT blue histogram, cKO$^{Kif5b}$ red histogram) (**h**); ***$P < 0.0001$ in a two-way ANOVA and Sidak test's correction for multiple comparison. **d–i** Graphs show mean ± S.E.M.

Ovalbumin-647, DQ-Ovalbumin, Alexa Fluor-647 and 488 Dextran 40,000 MW were obtained from Life Technologies. Nocodazole (used at a concentration of 10 μM) and latrunculin B (1 μM) were purchased from Sigma.

**Animal statement**. Housing and experiments were done as recommended by French regulations and experimental guidelines of the European Committee. The protocol was approved by the local animal care and use committee (CEE34, Paris, France; reference: APAFIS#18-079) under the number APAFIS#21213-2201804301641224v5.

**Mice**. VAV-Cre transgenic mice (Jackson Laboratory) were first crossed with $Kif5b^{+/-}$ mice to generate $Kif5b^{+/-};VAV-Cre$ animals. The latter were then bred with $Kif5b^{fl/fl}$ mice to generate $Kif5b^{fl/-};VAV-Cre$ (cKO$^{Kif5b}$) mutant mice and control (WT) $Kif5b^{fl/+}$ littermates. Mice were genotyped using a PCR with the primers P1 5′-TGAAGGCTAAGTCAGATATGGATGC-3′ and P3 5′-TTACTA ACTGAACCTGGCTTCCTAG-3′ to detect the floxed $Kif5b$ allele and P1 and P2 5′-GGATTGGCACCTTTACCTAGAAGG-3′ to detect the $Kif5b$ knockout allele. Transgenic OT-I TCR mice were bred with Rag1$^{-/-}$ mice to generate OT-I Rag1$^{-/-}$ mice. All mice used in experiments were between 8 and 12 weeks old. Mice were maintained under pathogen-free conditions and were handled in compliance with national and institutional guidelines.

**Quantitative PCRs**. cDNA was prepared from BMDC mRNA using Superscript II$^{™}$ reverse transcriptase and random primers (Invitrogen). Levels of $KIF5A$, $KIF5B$ and $KIF5C$ transcripts were determined in quantitative PCR (qPCR) assays using TaqMan Gene Expression Master Mix primers ($KIF5A$: Mm00515265_m1; $KIF5B$: Mm00515276_m1; $KIF5C$: Mm00500464_m1; Applied Biosystems). Each sample was amplified in triplicate in a real-time PCR cycler (ABI 7900) and then analysed with Sequence Detection Systems software (version 2.2.2, Applied Biosystems). The relative mRNA levels were quantified using the comparative Cτ method and normalised against the mean values of $ACTB$ as an endogenous control. Levels of $KIF5A$, $KIF5B$ and $KIF5C$ transcripts were then expressed as a proportion of the mean value for $KIF5B$ (arbitrarily set to 1 unit).

**Dendritic cell purification, and BMDC and cell line culture**. To purify DCs, spleens from wild-type or cKO$^{Kif5b}$ mice were digested with 500 μg/ml Liberase (Roche Diagnostics) and 50 ng/ml recombinant DNase I (Roche Diagnostics) in PBS. The DCs were pre-enriched from splenocytes using a very-low-density gradient. In brief, splenocytes were first resuspended in 4.2 mL of RPMI medium containing 5 mM EDTA, 5% FCS and 1 mL Optiprep (Sigma) and then loaded between 3 mL PBS containing 5 mM EDTA, 5% FCS and 1 mL Optiprep (bottom layer) and 1.8 mL of PBS containing 5 mM EDTA and 5% FCS (top layer). After 20 min of centrifugation at 1800 rpm at room temperature, the low-density fraction was collected at the interface between the top and middle layers. For the cross-presentation assay, the cells were immunostained with the following antibodies (all diluted 1:200): anti-CD11c BV711, anti-IA/IE FITC, anti-CD8α BV450, anti-CD11b PE, anti-F4/80 PE/Cy7, anti-GR1 APC, anti-CD19 PerCP, and anti-B220 BV650. Live cells (gate 1) were gated based on forward and side scatter, the doublets were excluded (gate 2), as well as B cell CD19$^+$ (gate 3), macrophages F4/80$^+$ (gate 4) and granulocytes Gr1$^+$ (gate 5). CD11c$^+$ IA/IE$^+$ population was selected (gate 6) to separate CD8α$^+$ DCs (gate 7) and CD11b$^+$ DCs (gate 8). The sorting was done on a BD FACS ARIA-II. For the others assays, the cells were immunostained with the following antibodies (all diluted 1:200): anti-CD19 PerCP/Cy5.5, anti-CD11c APC, anti-IA/IE FITC, anti-CD8α BV450, anti-CD11b APC-CY7, to identify CD11b DCs (CD19$^-$CD11c$^+$ IA/IE$^+$ CD11b$^+$), and CD8α DCs (CD19$^-$CD11c$^+$ IA/IE$^+$ CD8α$^+$) by flow cytometry. For the generation of

BMDCs, bone marrow was isolated from femurs of 8- to 12-week-old mice. Cells were then cultured in medium supplemented with GM-CSF, 15% heat-inactivated FCS, 1% non-essential amino acids, 1 mM sodium pyruvate, 100 U/ml penicillin and 100 U/ml streptomycin. After 8–10 days of culture, the BMDCs were used in experiments. Flow cytometry data analysis was performing using Flow Jo (Treestar) software version 10.

The B16-F10-OVA melanoma cell line was cultured at 37 °C in a 5% CO$_2$ atmosphere in DMEM, 10% FBS, 2 mM glutamine, 100 U/ml penicillin, 100 U/ml streptomycin, 50 μM β-mercaptoethanol, 25 mM HEPES, 1% non-essential amino acids and 1 mM sodium pyruvate.

**Cross-presentation assays**. For in vitro cross-presentation assays with sOVA (Worthington Biochemical Corporation), sorted cDCs or BMDCs were incubated with different concentrations of sOVA or different concentrations of the minimal peptide SIINFEKL as a control for cell-surface MHC class I. Six hours later, the cells were washed three times with PBS 0.5% BSA and co-cultured with OT-I T cells for 16 h. Secretion of IL-2 was then quantified using an ELISA. For cross-presentation with BMDCs, 50,000 cells per well were used. For cross-presentation with DCs purified from spleen, 20,000 cells per well were used. In assays with purified DCs and BMDCs, we used DC/OT-I T ratio of 1/3. For in vivo cross-presentation assays, naïve OT-I cells were purified from spleen and lymph nodes of OT-I Rag−/− mice and labelled with Violet Proliferation Dye 450, according to the manufacturer′s instructions. One to two million OT-I cells were then injected intravenously (iv) per mouse and the day after, mice were immunised iv with 500 ng of fusion protein containing OVA (P3UOVA) complexed with hamster anti-mouse CD11c monoclonal (clone N418) antibody (CD11c/P3UOVA). Three days after, mice were euthanized, the spleen was removed and processed for flow cytometry: splenocytes were stained with anti-CD19 PerCP/Cy5.5; anti-CD8α APC; anti-TCR Vα2 PE. OT-I cells were identified as live CD19$^-$CD8a$^+$Vα2 TCR$^+$ and violet proliferation dye V450$^+$ cells. Their activation was quantified by analysing the dilution of Violet Proliferation Dye V450 and by determining the division index of precursor cells.

**Ovalbumin bead cross-presentation**. OVA–BSA-coated beads were prepared by attaching different ratios of OVA:BSA concentrations (10 mg/mL OVA alone; 5 mg/mL OVA: 5 mg/mL BSA, 2.5 mg/mL OVA: 7.5 mg/mL BSA and 10 mg/mL BSA alone) to 3 μm latex beads by passive adsorption in PBS at 4 °C overnight. After several rounds of washing in PBS, OVA–BSA-coated latex beads were incubated with DCs for 6 h. Next, OT-I T cells were added to the culture for 16 h. IL-2 secretion was quantified using an ELISA.

**Vaccinia-ovalbumin**. BMDCs from WT or cKO$^{kif5b}$ mice were infected by vaccinia viruses expressing ovalbumin (OVA) or peptide SIINFEKL (S8L) or by the control strain WR1354 (obtained from ATCC) for 6 h at a multiplicity of infection (MOI) of 30, 10 and 3, respectively. Next, the cells were fixed with 1% for-maldehyde, washed successively with 0.2 M glycine pH 7.4, pH 7.0 and PBS, and added to transgenic TCR OT-I T cells seeded at $2 \times 10^5$/well in 96-well plates to give an effector:target (E:T) ratio of 2:1. IFN-γ secretion by OT-I cells was analysed after a 12-h incubation using a standard ELISA. The anti-IFN-γ capture mono-clonal antibody was clone R4-6A2, and the detection monoclonal antibody was clone XMG1.2 (both BD Pharmingen), both used at a concentration of 0.2 mg/50 ml/well. The incubation times were 5 h for the supernatant, 45 min for the detection monoclonal antibody, and 30 min for the streptavidin-HRP (Sigma). Tetramethylbenzidine was used as the substrate. H-2Kb-S8L complexes were detected by sequential incubation with 25D1.16 monoclonal antibody (10 μg/mL), FITC-labelled goat anti-mouse antibody (1:50; Biolegend), and Alexa488-labelled

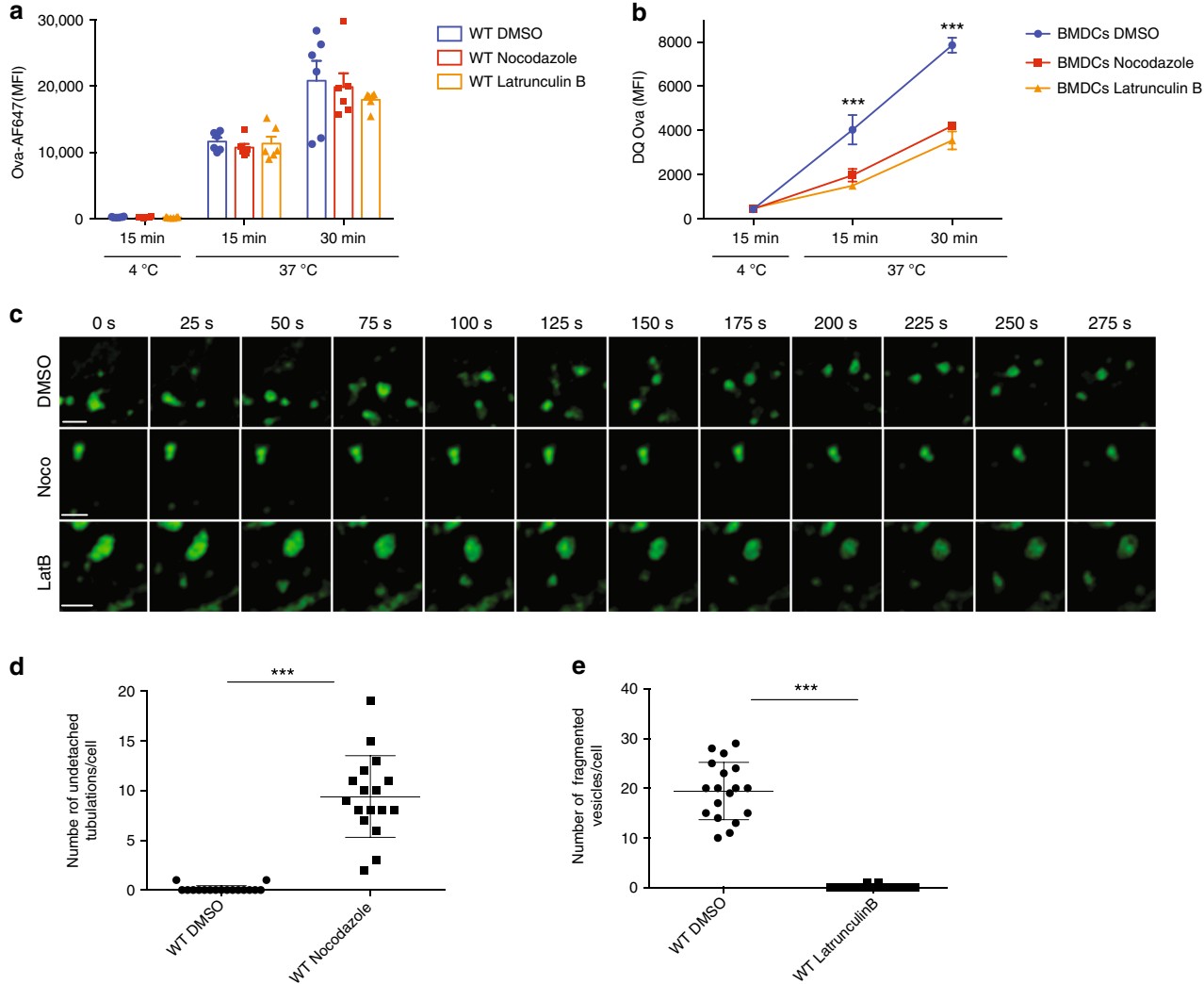

**Fig. 7 Microtubules are essential for tubule scission. a** BMDCs from WT mice pre-treated with DMSO (blue histogram or line), nocodazole (red histogram or line) or LatB (yellow histogram or line) were incubated for 15 min or 30 min at 37 °C or for 15 min at 4 °C with Alexa-Flour-647-OVA, prior to flow cytometry analysis. Graphs are representative of six independent experiments. **b** BMDCs from WT mice pre-treated with DMSO, nocodazole or LatB were incubated for 15 min and 30 min at 37 °C with DQ-OVA. The cells were then washed. DQ-OVA degradation was analysed by flow cytometry after different chase periods. Graphs are representative of six independent experiments. Statistical analysis: ***$P < 0.0001$ in a two-way ANOVA and Sidak test's correction for multiple comparison. **c** Spinning disk video microscopy of BMDCs from WT mice pre-treated with DMSO, nocodazole or LatB, labelled with WGA-488 and plated on glass coverslips. Representative series of images are shown every 25 s. Bars: 2 μm. See also Supplementary Movies 3, 4 and 5. **d** The number of tubulations unable to detach per cell pre-treated with DMSO or nocodazole, observed for 5 min with spinning disk microscopy ($n = 18$ cells per condition). Statistical analysis: ***$P < 0.0001$ in a two-tailed unpaired Student's $t$ test. **e** The number of attached tubulations per cell pre-treated with DMSO or LatB, observed for 5 minutes with spinning disk microscopy ($n = 17$ cells per condition). Statistical analysis: ***$P < 0.0001$ in a two-tailed unpaired Student's $t$ test. **a**–**e** Graphs show mean ± S.E.M.

goat anti-FITC antibody (1:100; Invitrogen). Fluorescence was measured on a BD LSRFortessa analyzer.

**OT-II presentation assay**. For the OT-II presentation assay (Worthington Biochemical Corporation), sorted cDCs or BMDCs were incubated with different concentrations of sOVA. Six hours later, cells were washed three times with PBS 0.5% BSA and co-cultured with OT-II T cells for 16 h. IL-2 secretion was then quantified using an ELISA. For in vivo assays measuring antigen presentation assays to CD4+ T lymphocytes, naïve OT-II cells were purified from the spleen and lymph nodes of OT-II Rag−/− mice and labelled with Violet Proliferation Dye, according to manufacturer's instructions. One to two million OT-I cells were then injected iv per mouse and the day after, mice were immunised iv with 200 ng CD11c/P3UOVA. Three days after, mice were euthanized, the spleen was removed and processed for flow cytometry: splenocytes were stained with anti-CD19 PerCP/Cy5.5; anti-CD4 FITC and anti-TCR Vb5.1/5.2 PE. OT-II cells were identified as live CD19−CD4+Vβ5.1/5.2 TCR+ and Violet proliferation dye V450+ cells. Their activation was quantified by analysing the dilution of Violet Proliferation Dye V450 and by determining the division index of precursor cells.

**In vivo tumour growth**. On day 0, WT and cKO$^{Kif5b}$ mice were injected subcutaneously with $0.2 \times 10^6$ B16-OVA cells. On day 1, both groups of animals were injected (or not) with $2.5 \times 10^6$ transgenic OT-I cells (bearing an OVA-specific TCR). The OT-I CD8+ T cells used in adoptive transfer had been prepared from the lymph nodes of OT-I mice by negative depletion. Growing, encapsulated tumours were clearly visible on day 7. Tumour growth was measured daily (using callipers) until day 25. Mice were killed when the tumours ulcerated and/or had a mean diameter of 15 mm. The tumour volume was determined as $V = 4/3 \times \pi \times R1 \times R2 \times R3$.

**Antigen internalisation assay**. The endocytic activity of BMDCs or cDCs was assessed using OVA Alexa Fluor 647 conjugate (Thermo Fisher Scientific) (5 μg/ml for BMDCs, and 150 μg/ml for cDCs) for 15 min, 30 min and 1 h. Cells were washed with PBS 1% BSA and resuspended in 100 μl of FACS buffer (PBS containing 2% FCS and 1% BSA). OVA uptake was monitored by flow cytometry.

**Antigen degradation assay**. A self-quenched conjugate of ovalbumin that fluoresces upon proteolytic degradation (DQ-OVA, Invitrogen) was used to measure Ag degradation by DCs. cDCs and BMDCs ($1 \times 10^6$) were suspended in

400 μl of complete medium containing 150 μg/ml (for cDCs) or 5 μg/ml (for BMDCs) DQ-OVA, pulsed at 4 °C or 37 °C for 30 min (for cDCs) or 15 min (for BMDCs), and then chased for the indicated times. Cells were washed with PBS 1% BSA and resuspended in 100 μl of the FACS buffer described above. The mean fluorescence intensity (MFI) of DQ-OVA in the DCs was measured by flow cytometry.

**NK cell degranulation**. Splenic NK cells were purified using the NK cell isolation kit (Miltenyi Biotech). Purified cells were cultured at $2 \times 10^5$/mL in RPMI with 10% FCS and 30 ng/mL recombinant murine IL-15 (Peprotech) for 3 days. NK cells were then mixed in 96-well plate with YAC-1 cells (Ratio 1 NK cell: 3 YAC-1 cells) and with PE anti-mouse CD107a (BioLegend). After 3 h culture, cells were stained with APC anti-mouse NK1.1 (BD Pharmingen) and CD107a expression in NK cells was quantified by flow cytometry.

**Flow cytometry-based phagocytosis assay**. In all, 3 μm NH2 beads (Polysciences) were activated with 2 mg/mL sulfo-NHS-LC-Biotin, washed with PBS-glycine 1 M, and stained with streptavidin Alexa Fluor 488. The beads were then added at different ratios to the BMDCs for 15 min at 37 °C. Immediately before FACS analysis, cells were washed and resuspended in pH 4.0 buffer containing 0.2 mg/mL Trypan Blue and 1 M citrate. The percentage of fluorescence represents only internalised beads and not those bound to the cell surface.

**Intraphagosomal degradation assay**. OVA (0.5 mg/mL) was coupled covalently after the activation of 3 μm latex beads with glutaraldehyde 8%. The BMDCs were pulse-chased with coupled latex beads at the indicated times. To collect the latex beads, the BMDCs were disrupted in lysis buffer and centrifuged at $150 \times g$ for 4 min at 4 °C. Rabbit polyclonal anti-OVA and FITC-coupled anti-rabbit antibodies were used to label the latex beads. The mean fluorescence intensity (MFI) of OVA-latex beads was analysed using flow cytometry.

**Endosomal pH measurement**. Dendritic cells were pulsed with a mixture of 40 kDa dextran fluorescein (1 mg/ml) and 40 kDa dextran Alexa-647 (1 mg/ml) (Molecular Probes) for 10 min in a water bath at 37 °C and then extensively washed with cold PBS 1% BSA. The cells were resuspended in complete medium incubated at 37 °C (for "chasing") for the indicated times and immediately placed on ice to stop the reaction. To determine the endocytic population, a control experiment was maintained at 4 °C during the pulse. Cells were washed with PBS 1% BSA, resuspended in 100 μl of FACS buffer, and analysed using FACS. An FL1/FL4 gate was used to select cells that had endocytosed the fluorescent probes. Values were compared with a standard curve obtained by resuspending the cells that had endocytosed dextran at fixed pH (ranging from pH 5 to 8) in buffer containing 0.1% Triton X-100 for 8 min. The cells were immediately analysed using FACS to determine the emission ratio of the two fluorescent probes at each pH.

**Protease activity assay**. Protease activity assays were performed on Mithras LB 940 microplate reader by measuring the release of fluorescent N-Acetyl-Methyl-Coumarin in citrate buffer (pH 5.5) at 37 °C. Specific substrates for CatB/L (Z-Phe-Arg-NHMec) were from Bachem.

**Recycling assay**. Purified spleen $CD8\alpha^+$ and $CD11b^+$ DCs and BMDCs from WT or $cKO^{kif5b}$ mice were surface-labelled with either FITC-coupled anti-H-2K$^b$ or Alexa-488-coupled transferrin receptor for 30 min at 4 °C. After washing with PBS 1% BSA, cells were incubated for 30 min at 37 °C in complete medium to allow internalisation. DCs were then resuspended in 0.5 M NaCl 0.5% acetic acid, pH 3 buffer (stripping solution) for 10 min on ice. Next, washed cells in cold PBS were resuspended in complete medium and incubated at 37 °C for the indicated time period to allow recycling of MHC-I and TrfR to the cell surface. After each time period, DCs were resuspended in stripping solution for 10 min on ice. Cells were then fixed in 1% PFA. The MFI of FITC or Alexa 488 was measured using FACS. The percentage of recycled molecules was measured using the equation $(T0-Tx)/T0 \times 100$, where T0 was the MFI of cells at 0 min, and Tx was the MFI of cells at the indicated times. The recycling assay with primaquine was performed on BMDCs from WT or $cKO^{kif5b}$ mice. BMDCs were first incubated with biotinylated anti-H-2K$^b$ antibody at 4 °C for 30 min. After thorough washing, cells were incubated at 37 °C for 30 min in the presence of 220 μM primaquine. Next the cells were stripped for 10 min at 4 °C. After washing, cells were incubated at 37 °C and removed at various intervals. Recycled H-2K$^b$/Ab complexes were detected by flow cytometry using streptavidin Alexa-488 conjugate.

**Immunofluorescence**. Bone marrow-derived DCs were plated on glass coverslips and fixed by incubation for 15 min on ice in 3.7% w/v paraformaldehyde followed by quenching for 10 min with 50 mM NH₄Cl in PBS. The cells were incubated for 1 h with specific primary antibodies in permeabilization buffer (PBS with 1 mg/mL BSA and 0.05% w/v saponin (Sigma-Aldrich)), washed twice, and incubated for another hour with the fluorescently labelled secondary antibody in permeabilization buffer. Lastly, the cells were mounted on slides in Prolong Gold antifade reagent in the presence of DAPI (Invitrogen Carlsbad, CA). Confocal microscopy

was performed with a Zeiss LSM 700 system (Carl Zeiss) and a 63x NA1.4 objective. Images were processed with the Zeiss LSM Image Browser (Carl Zeiss) and ImageJ software (version 1.43).

**Spinning disk video microscopy**. High-speed video microscopy of early endosome dynamics in WGA-labelled BMDCs was performed using a Zeiss Axio Observer Z1 equipped with Yokogawa spinning disk technology and a 63x NA1.46 objective. The WGA-labelled BMDCs were plated on glass coverslips and the maintained at 37 °C in a 5% $CO_2$ atmosphere. Spinning disk images were then acquired for 5 min (exposure time: 50 ms). Image sets were processed with Fiji software.

**Quantification of attached tubulations**. We monitored the detachment of tubulations (~1 μm in size) from vesicular structures labelled with WGA-488 (pulsed for 5 min and chased for 5 min) over a 5 min period. Attached tubulations were defined as tubulations identified during the first minute of the acquisition and that did not detach during the rest of the acquisition.

**Quantification of fragmented vesicles**. We monitored the vesicle individualisation from vesicular structures labelled with WGA-488 (pulsed for 5 min and chased for 5 min) over a 5-min period. A fragmented vesicle was defined as a vesicle that detached from the WGA-labelled endosome during the acquisition.

**Statistical analysis**. All analyses of statistical significance were performed with GraphPad Prism software (version 7). All P values < 0.05 were considered significant. Of note, *, ** and *** indicate P values < 0.05, 0.005 and 0.0001, respectively. The exact P value for each experiment is reported in the Source Data file.

**Reporting summary**. Further information on research design is available in the Nature Research Reporting Summary linked to this article.

## Data availability

The main data supporting the finding of this study are available within the Article and its Supplementary Information. All other relevant data are provided in the "source data file", and available from the corresponding author on request.

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

## Acknowledgements

We thank the staff at Imagine Institute's animal facility for their assistance and especially Jade Banlier. We thank Louison Lallement for her technical assistance with the spinning disk microscopy, Sebastian Montealegre for his technical assistance in DC functional assays and Radia Aïssani-El Fertas for her technical assistance with the anti-tumour response experiments. This work was funded by INSERM, the ARC foundation (PJA 20161204628; PJA 20181207755), and La Ligue Contre le Cancer (JML/NF-RS17/75-2; RS19/75-1; RS20/75-1). M.B. received a doctoral fellowship from the Ministère de l'Education Nationale de la Recherche et de la Technologie and Imagine Thesis Award (2019). P.V.E. was funded by ANR-14-CE11-0014-01 and FRM DEQ20130326539. This work was supported by State funding from Agence Nationale de la Recherche under "Investissement d'Avenir" programme (grant: ANR-10-IAHU-01).

## Author contributions

M.B., F.X.M., M.K. and F.E.S. designed and performed experiments, analysed data. G.M. designed experiments, analysed data and wrote the paper. M.B., F.X.M., M.K., S.M., N.G., J.-D.H., A.F., P.V.E., G.d.S.B., F.E.S. and G.M. revised the manuscript for critical content.

## Competing interests

The authors declare no competing interests.
