## [Peer Review File · Nature Communications]

Reviewers' comments:

Reviewer #1, expert in cell biology and antigen presentation (Remarks to the Author):

The report by Belabed et al describes a new role for the motor protein kinesin-1 by regulating endosomal maturation. Kinesin-1 is a motor protein, which moves along the microtubules using ATP hydrolysis and transport cargo in the cells. The authors propose that kinesin-1 (Kif5b) is responsible for the scission of endosomal tubulation allowing the maturation of early endosomes (EEA1+) into recycling (Rab11+) or late endosomes (LAMP1+), a process required for antigen cross presentation in dendritic cells (Figure 7F). This work is interesting and new but some mechanistic is lacking. For example Kinesin 1C has been involved in the transport of Golgi apparatus to ER (Dorners et al, JBC 1998) and in Golgi tethering (Lee et al, eLife 2015). The authors should quantify the number of EEA1, Rab11 and LAMP1 positive vesicles at the steady state as well as monitor the morphology of ER and Golgi apparatus in wt and Kif5b deficient DCs. Mechanistically does Kif5b binds EEA1? What is the percentage of Kif5b-EEA1 positive vesicles (not very high Figure 6B)? If Kif5b recruits EEA1+ endosomes and promotes the scission of EEA1+ endosomes to Rab11+ and LAMP1+ compartments, why is MHCII antigen presentation not affected and why do we see more colocalization of OVAs in LAMP1+ lysosomes in Kif5b deficient DCs at 60 min in comparison to control cells (Figure 4)? The rate of MHC I recycling molecules is very high especially in CD8+ DCs. Usually, MHC recycling is very efficient and difficult to assess. The use of primaquine, which slow the return of receptors on the cell surface by increasing their intracellular pool makes possible to measure recycling (Reid PA et al, Nature 1990). Can the authors explain why they have such high rate of MHC I recycling without using primaquine? How many cells were used for this experiment? Did the authors look at the recruitment of the V-ATPase and NADPH oxidase in Kif5b deficient endosomes? Is the recruitment of LRO to early endosomes also impaired in Kif5b deficient DCs? Did they monitor proteases activities in Kif5b deficient endosomes?

Minor points:

Figure 1B: gating strategy should be described

Figure 1D: It is surprising that BMDCs are better cells to cross present OVAs than CD11b and even CD8+ DCs. The uptake of OVAs seems to be the same between BMDCs and CD8+ DCs (Figure 2A). What is the number of DCs used per well to perform this assay? The ratio DC: T cells? It is not describe in the Figure legend or in the Mat and Methods section.

Figure 3C: statistics should be done

Overall the IF figures seem a bit overexposed (especially Figure 4). Will be nice to see the edge of the cells.

Reviewer #2, expert in cross-presentation (Remarks to the Author):

In their manuscript, the authors investigate the influence of Kif5b on cross-presentation and endosome trafficking. They use conditional Kif5b KO mice to demonstrate that Kif5b deficiency inhibits antigen degradation, endosome acidification and maturation and cross-presentation. Additionally, they demonstrate that early endosomes from Kif5b deficient cells are less motile and show impaired scission of tubules.

The manuscript is very clear and describes a novel role of Kif5b in cross-presentation, which is an interesting subject to the field. The experiments are very convincing and depicted in clear figures. The manuscript is already of high quality.

Nevertheless, I have the following remarks:

1) on page 6, the authors show that endosomes from Kif5b-deficient DCs display impaired

acidification. This is misleading. Of course, impaired endosome acidification can be the cause of changes in pH. However, it could also be that the dextrane used does not reach acidic compartments because of impaired lysosome trafficking. This should be clarified in the text.

2) Top of page 7: here, the authors claim that they investigate the effect of Kif5b on the vacuolar pathway of cross-presentation. This is not true and misleading. In their experiments, the authors did not distinguish between the vacuolar pathway and the endosome-to-cytosol pathway at all. Given the effect of Kif5b deficiency on endosome trafficking, it is very likely to have an effect on cross-presentation, regardless whether the antigens are cross-presented via the vacuolar or via the endosome-to-cytosol pathway. This also should be clarified in the text.

3) Middle of page 7: here, the authors describe a lower percentage of colocalization between OVA and LAMP1. However, Fig 4 clearly demonstrates that the colocalization is not lower but rather delayed. This should be reworded.

4) middle of page 9: here they state as a sort of conclusion that "Kif5b binds to EEA1-positive endosomes..... and allows subsequent sorting of Ag/MHC-I complexes". This should be reworded since they didn't prove any of these statements and this reflects overinterpretation of the results.

5) Top of page 11, they state that "the cross-presentation defect in the absence of Kif5b resulted from an impairment in early endosome dynamics and membrane fission." Also this is not proved in the manuscript and reflects overinterpretation. Should be reworded.

6) The authors showed several effects of Kif5b deficiency: antigen degradation, endosome acidification, endosome maturation, reduction in MHC-I recycling, antigen trafficking towards lysosomes and tubulation scission. But which of these effects now are responsible for impaired antigen presentation? As the authors point out correctly in their discussion, it is generally accepted that rapid antigen degradation prevents cross-presentation, meaning that impaired antigen degradation and trafficking towards lysosomes might not be critically involved in inhibition of cross-presentation. Alternatively, if peptides derived from soluble antigens are loaded onto MHC I in endosomes (regardless of vacuolar or endosome-to-cytosol pathway), they need to be transported towards the cell membrane, which presumably occurs via Rab11+ endosomes. Since the authors show that Kif5b deficiency prevents such protein transport, this might be the main reason for impaired cross-presentation. The discussion on how Kif5b might influence cross-presentation should be extended in the manuscript in general.

Sven Burgdorf

Reviewer #3, expert in innate instruction of adaptive immunity (Remarks to the Author):

This manuscript investigates kinesin-1 in cross-presentation. This is not the first time that kinesin-1 has been reported to play a role in DC function and cross-presentation (this should be cited in the Introduction) and/or receptor recycling. Here, the authors investigate the response in more detail. Overall the manuscript lacks rigour and coherence.

A major shortfall of the manuscript is that there is an overinterpretation of a specific role for kinesin-1 in cross-presentation. The manuscript lacks in depth analysis of other forms of antigen presentation. The role of kinesin-1 in MHC II antigen presentation and direct MHC I antigen presentation needs to be clarified in a more rigorous manner. Indeed it is surprising that MHC II antigen presentation is not affected given the significant impact of kinesin-1 KO on endosomal biology and antigen proteolysis. Likewise, direct MHCI presentation may be altered due to the role of kinesin-1 in MHCI recycling. The supplementary data in SFig4 suggests there is a defect in MHCII presentation of soluble OVA by CD8 DC. Moreover, the manuscript requires clarification regarding the different forms of antigen used.

Bead-OVA antigen presentation data should be included. The data for cross-presentation contradicts the accepted view that cDC1 are the more efficient cross-presenting cell type. For example, published data suggests there is little cross-presentation of soluble OVA by CD11b+ DC eg. Schnorrer. 2006. How do the authors explain this discrepancy between their cross-presentation outcomes and those previously reported? The link between cross-presentation of soluble antigen in vitro (Fig 1) and cross-presentation of tumour associated antigen in vivo (Fig 2) is not clear, given these are two very different types of antigen.

Tumour data is used as evidence for a role of kinesin-1 in cross-presentation in vivo. While there is clearly an impact of kinesin-1 on the growth kinetics of the tumour, this also occurs in the absence of OT-I transfer suggesting mechanisms other than OVA cross-presentation could be playing a role. Are other immune populations eg CD8 and CD4 T cells, B cells and NK cells functional and present in normal numbers in the kinesin-1 cKO? Concluding the changes in tumour growth and/or mouse survival are solely due to cross-presentation is an overinterpretation of the data. To measure cross-presentation in vivo, the authors need to perform in vivo transfer of OTI and OTII cells and measure proliferation in response to different forms of (cross-presented) antigen.

In Figure 5, analysis of OVA trafficking shows higher colocalization with EEA-1 and less with LAMP and Rab11 early in the cKO but at later time points this is enhanced compared to WT. How do the authors explain this? Do these experiments take into account that in the WT there is significantly more antigen proteolysis compared to cKO?

It is unclear how the data in Figure 7 directly relates to kinesin-1. Given that tubule formation requires actin polymerisation then it is not surprising endosomal tubulations would not be present in cells without tubules. It is unclear how this data fits into the theme of the manuscript.

The Discussion is poorly written and needs to be more concise.

Throughout the manuscript there are many statements that over-interpret the data presented:
"Our results provide the first genetically based evidence in support of this hypothesis, and highlight kinesin-1 as a critical regulator of Ag degradation and Ag sorting from the early endosomes."
"Our results highlight kinesin-1's newly recognized role as a molecular checkpoint that modulates the balance between antigen degradation and cross-presentation."
"Our results show that kinesin-1 (i) has an essential role in the Ag and MHC-I endocytic trafficking upstream of cross-presentation"
"Taken as a whole, these results suggest that the defect in T cell priming observed in the absence of kinesin-1 was specific to cross-presentation." "In contrast, the presentation of endogenous Ag to MHC-I molecules and the presentation of exogenous Ag to MHC-II molecules in cKOf5b mice was unaffected".
"Taken as a whole, these data indicate that Kif5b binds to EEA1-positive sorting endosomes, drives the scission of tubular structures, enables vesicles to mature into recycling or late endosomes, and thus allows the subsequent sorting of Ag/MHC-I complexes"

The majority of the statistical analysis of data is performed using an unpaired t test which is not the correct analysis for the type of data presented.

Minor Comments

More detail is required to explain Fig 1A in the text, how was Kif5b measured, what are Kif5c, Kif5a?

Fig 2C. Mouse tumour images are unnecessary.

Figure panels for Figure 3 should be ordered so CD8 DC are aligned. Have technical repeats been performed for endosomal pH measurements?

Reviewer #4, expert in cell biology and antigen presentation (Remarks to the Author):

The authors show here that in absence of Kif5b expression, cross presentation in DCs is impaired and anti tumor immune responses are reduced. They also show that endocytic degradation, acidification, recycling and cargo transfer to late compartments are all impaired.

The results are clearcut and the experience are well designed. The authors conclude that these defects in intracellular traffic cause reduced antigen cross presentation and impair anti tumor immune responses. They propose that Kif5b plays a critical role in the control of the balance between antigen degradation and cross presentation.

The main problem with this conclusion is that, based on previous studies (cited by the authors), the defects in endocytic functions and intracellular traffic in Kif5b KO dendritic cells would suggest increased and not impaired, antigen cross presentation. Several groups have shown that reduced degradation, acidification and lysosomal transport are all associated with better cross presentation. The main difference with the Kif5b KO is that recycling is also impaired. It is therefore most likely that impaired cross presentation in Kif5b is in fact due, mainly, to defective MHC I re-cycling. Supporting this possibility with experimental data would strengthen the paper. What are the effects of nocodazole and latrunculin B on MHC class I re-cycling? And in all cases this issue should be better discussed.

Other remarks:

- In the tumor growth experiment it is important to monitor the T cell responses (eventually using a model tumor antigen).
- In all figures showing different types of DCs, please keep the same order for the different DC-types.
- The summary scheme in Figure 7F is not very helpful... If kinesin 1 (does this mean Kif5b?) is involved in both recycling and degradation (as the results show), it is not possible to know if the effect on cross presentation is due to either or both...

In response to Reviewer #1

The report by Belabed et al describes a new role for the motor protein kinesin-1 by regulating endosomal maturation. Kinesin-1 is a motor protein, which moves along the microtubules using ATP hydrolysis and transport cargo in the cells. The authors propose that kinesin-1 (Kif5b) is responsible for the scission of endosomal tubulation allowing the maturation of early endosomes (EEA1+) into recycling (Rab11+) or late endosomes (LAMP1+), a process required for antigen cross presentation in dendritic cells (Figure 7F). This work is interesting and new but some mechanistic is lacking. For example Kinesin 1C has been involved in the transport of Golgi apparatus to ER (Dorners et al, JBC 1998) and in Golgi tethering (Lee et al, eLife 2015). **The authors should quantify the number of EEA1, Rab11 and LAMP1 positive vesicles at the steady state as well as monitor the morphology of ER and Golgi apparatus in wt and Kif5b deficient DCs.**

We thank the reviewer for raising this point. As required by the reviewer, we have performed new experiments to assess the consequences of the absence of Kif5b on the number of vesicles and morphology of the different endosomal compartments, Golgi apparatus and ER. We found similar number and morphology of EEA1, Rab11, Lamp1, calnexin and golgin-97-positive compartments in Kif5b-deficient or -proficient BMDCs, indicating that Kif5b is not involved in maintaining the integrity of these compartments. This new result is now included as supplemental data in Figure S9 and in the text Page 8.

Mechanistically does Kif5b bind EEA1?

To address this question, we over expressed Kif5b or KLC1 in Hek293T cell line. Neither Kif5b nor KLC1 were able to immunoprecipitate endogenously expressed EEA1. We choose to not include this negative result in the manuscript.

Fig: Hek293T cells were untransfected (-) or transfected with either GFP, GFP-KIF5B or GFP-KLC1. Lysates were then immunoprecipitated with a control isotype (Santa cruz biotechnology) or with an anti-GFP monoclonal antibody (Roche) and immunoblotted with anti-KIF5B (Proteintech Europe), anti-KLC1 (Abcam) or anti-EEA1 (BD). When cellular lysates were subjected to immunoprecipitation with anti-GFP, we found that GFP-KIF5B was co-

immunoprecipitated with endogenous KLC1 and GFP-KLC1 was co-immunoprecipitated with endogenous KIF5B. However, EEA1 was not co-immunoprecipitated neither with GFP-KIF5B nor with GFP-KLC1.

What is the percentage of Kif5b-EEA1 positive vesicles (not very high Figure 6B)?

The quantification of the colocalization between EEA1 and Kif5b has been done on 20 cells (EEA1/Kif5b overlap: $53.7\% \pm 7.7\%$ (Fig. 6B) Page 9. To have a better representation of the colocalization, we have now added the intersection between both stainings in withe in Fig6B.

If Kif5b recruits EEA1+ endosomes and promotes the scission of EEA1+ endosomes to Rab11+ and LAMP1+ compartments, why is MHCII antigen presentation not affected and why do we see more colocalization of OVAs in LAMP1+ lysosomes in Kif5b deficient DCs at 60 min in comparison to control cells (Figure 4)?

In order to further evaluate the role of Kif5b in MHC-II antigen presentation, we have now studied the invariant chain Ii degradation as Ii is well known to have a key role in MHC-II trafficking and peptide loading. WT and cKO^{Kif5b} BMDCs were analysed by western blotting for the presence of Ii and its intermediate degradation products. The Ii degradation was not altered in Kif5b-deficient BMDCs. This result is now added to Fig. S5B and in the text Page 5-6.

We then investigated the *in vivo* proliferation of injected OT-II CD4+ T cell (labelled by violet proliferation dye) in WT and cKO^{Kif5b} mice that were primed the next day with fusion protein containing OVA (P3UOVA), complexed with hamster anti-mouse CD11c antibody (CD11c/P3UOVA) to target specifically CD11c⁺ DCs (Kratzer R JI 2010). Confirming our previous *in vitro* data (Fig.S5A), we did not observe different proliferation ratio between OT-II CD4+ T cells injected in WT or cKO^{Kif5b} mice. These new results are now included as supplemental data in Figure S5C and in the text Page 6 to strengthen that Kif5b does not play a role in MHC-II antigen presentation.

To explain the absence of Kif5b detectable role in MHC-II antigen presentation, we can speculate that the Ag degradation defect (which is not totally but only partially impaired and delayed) in Kif5b-deficient DCs does not affect with the same severity the MHC-II pathway. Furthermore, we cannot rule out the possibility that other member(s) of the kinesin family may regulate MHC-II trafficking or compensate for the loss of kinesin-1. Finally, we cannot exclude that the Ag degradation defect will not be directly responsible for the Ag cross-presentation defect and thus will not affect MHC-II Ag presentation. Indeed, it is possible that the Ag cross-presentation defect mainly depends of the defective MHC-I trafficking and recycling and thus will not impact MHC-II presentation. We have now discussed this point in the P14 of the discussion section of the revised manuscript.

Indeed, as raised by the reviewer “we observed more colocalization of OVAs in LAMP1+ lysosomes in Kif5b deficient DCs at 60 min in comparison to control cells (Fig4)”. We explain this discrepancy by the defect of soluble ovalbumin degradation observed in Kif5b-deficient BMDCs compared to WT BMDCs (Fig3B) and the delay in recruitment of early endosomes to late endosomes/lysosomes also observed in Kif5b-deficient BMDCs compared to WT BMDCs (Fig6H, 6I). We have now clarified this point in the text Page 8.

The rate of MHCI recycling molecules is very high especially in CD8+ DCs. Usually, MHC recycling is very efficient and difficult to assess. The use of primaquine, which slow the return of receptors on the cell surface by increasing their intracellular pool makes possible to measure recycling (Reid PA et al, Nature 1990). Can the authors explain why they have such high rate of MHCI recycling without using primaquine? How many cells were used for this experiment?

We thank the reviewer for raising this point. The acid stripping approach is commonly used in cell biology (Brian et al. Journal of Vir 2007; Finetti et al. JCS 2014; Osborne et al. JI 2015; Montagnac et al. Cur Bio 2011; Delevoye et al. Cell Rep 2014) to assess molecule recycling. In DC, this method was more recently used by Cebrian et al. 2016 to assess the role of Rab22 on MHC-I recycling. For this approach, we used 1 million BMDCs per experimented point to assess the recycling of MHC-I molecules. As proposed by the reviewer, to confirm the defect of MHC-I recycling in Kif5b-deficient BMDCs, we performed the same assay in the presence of primaquine, known to increase the intracellular pool size of recycling molecules. In this setting, we were able to reproduce the defect of MHC-I recycling in Kif5b-deficient BMDCs. We have added this result in figure 5D and in the text Page 8-9.

Did the authors look at the recruitment of the V-ATPase and NADPH oxidase in Kif5b deficient endosomes?

We did not look at the recruitment of the V-ATPase and NADPH oxidase in Kif5b-deficient endosomes. This would be done in future study.

Is the recruitment of LRO to early endosomes also impaired in Kif5b deficient DCs?

We apologize if this point in our original manuscript was unclear. We have tested the recruitment of early endosomes to late endosomes/lysosomes by using Alexa-Flour-488-WGA to label early endosomes and Alexa-Flour-555-WGA to label late endosomes/lysosomes. Recruitment of early endosomes to late endosomes/lysosomes was monitored by acquiring live cell spinning disk microscopy images every 5 min over 120 min. The recruitment appeared to be slower in Kif5b-deficient BMDCs than in WT BMDCs (Fig. 6H, 6I). This result suggests that the recruitment of early endosomes to LRO (lysosome related organelle) or LRO to early endosomes is delayed in Kif5b-deficient DCs.

Did they monitor proteases activities in Kif5b deficient endosomes?

We thank the reviewer for raising this point on proteases activities in Kif5b-deficient endosomes. We have assessed the proteolytic activity of catB/L using total cell lysate. We have shown that the proteolytic activity of catB/L in the total lysate of Kif5b-deficient BMDCs was markedly decreased at 20 and 120min (New figure 3E and text Page 7-8) compared to WT BMDCs.

Minor points:

Figure 1B: gating strategy should be described

Following reviewer suggestion, we have now rewritten on page 16 the gating strategy to isolate purified DCs from the spleen.

Figure 1D: It is surprising that BMDCs are better cells to cross present OVAs than CD11b and even CD8+ DCs. The uptake of OVAs seems to be the same between BMDCs and CD8+ DCs (Figure 2A). What is the number of DCs used per well to perform this assay? The ratio DC: T cells? It is not describe in the Figure legend or in the Mat and Methods section.

Since the number of cells and quantity of antigen used in these different experiments were not the same, the capacity of cross-presentation between purified DCs and BMDCs cannot be compared. For cross-presentation with BMDCs, 50,000 cells per well were used with a ratio of 1 BMDC for 3 OT-I T cells. For cross-presentation with purified DCs from the spleen, 20,000 cells per well were used with a ratio of 1DC for 3 OT-I T cells. We have now clarified this experimental procedure in the revised version of the materiel and method section Page 17.

Figure 3C: statistics should be done.

We have now added statistical analysis for the pH experiment (new figure 3D).

Overall the IF figures seem a bit overexposed (especially Figure 4). Will be nice to see the edge of the cells.

The setup of the microscope in the different experiments was made in order to avoid overexposure signal. However, in the revised version of the manuscript we have tried to reduce in post treatment (Image J) the intensity of the staining. We have also added for Fig4, Fig5 and Fig6B the contour of the cells.

In response to Reviewer #2

In their manuscript, the authors investigate the influence of Kif5b on cross-presentation and endosome trafficking. They use conditional Kif5b KO mice to demonstrate that Kif5b deficiency inhibits antigen degradation, endosome acidification and maturation and cross-presentation. Additionally, they demonstrate that early endosomes from Kif5b deficient cells are less motile and show impaired scission of tubules.

The manuscript is very clear and describes a novel role of Kif5b in cross-presentation, which is an interesting subject to the field. The experiments are very convincing and depicted in clear figures. The manuscript is already of high quality.

We thank the reviewer for his comments and a detailed response to each remark raised is provided below.

Nevertheless, I have the following remarks:

1) on page 6, the authors show that endosomes from Kif5b-deficient DCs display impaired acidification. This is misleading. Of course, impaired endosome acidification can be the cause of changes in pH. However, it could also be that the dextrane used does not reach acidic compartments because of impaired lysosome trafficking. This should be clarified in the text.

The reviewer has raised a very interesting point. However, if we were facing a defect in dextran trafficking we should observe a pH defect also in CD8a+ DCs. To address the consequences of different endosomal pH between Kif5b-deficient BMDCs and WT BMDCs, we have assessed the activity of cathepsins B and L (catB/L). In the revised version of the manuscript, we show in Fig3E and pages 7-8, that the proteolytic activity of catB/L in the total lysate of Kif5b-deficient BMDCs was markedly decreased at 20 and 120min. These results strongly suggest that the change in the endosomal pH in Kif5b-deficient DCs can modify the activity of endosomal proteases and suggest a defect in Ag degradation as a result.

2) Top of page 7: here, the authors claim that they investigate the effect of Kif5b on the vacuolar pathway of cross-presentation. This is not true and misleading. In their experiments, the authors did not distinguish between the vacuolar pathway and the endosome-to-cytosol pathway at all. Given the effect of Kif5b deficiency on endosome trafficking, it is very likely to have an effect on cross-presentation, regardless whether the antigens are cross-presented via the vacuolar or via the endosome-to-cytosol pathway. This also should be clarified in the text.

Following reviewer's comment, we have now clarified this issue in the text Page 7.

3) Middle of page 7: here, the authors describe a lower percentage of colocalization between OVA and LAMP1. However, Fig 4 clearly demonstrates that the colocalization is not lower but rather delayed. This should be reworded.

We agree with the reviewer's comment that the OVA recruitment in LAMP1+ compartment is not lower but delayed. We have now clarified this issue in the text Page 8.

4) middle of page 9: here they state as a sort of conclusion that "Kif5b binds to EEA1-positive endosomes... and allows subsequent sorting of Ag/MHC-I complexes". This should be reworded since they didn't prove any of these statements and this reflects overinterpretation of the results.

Following reviewer's comments, we have now reformulated the text of page 10 in the new version of the manuscript.

5) Top of page 11, they state that "the cross-presentation defect in the absence of Kif5b resulted from an impairment in early endosome dynamics and membrane fission." Also this is not proved in the manuscript and reflects overinterpretation. Should be reworded. We have now reformulated the text of page 12 in the new version of the manuscript, according to reviewer comment.

6) The authors showed several effects of Kif5b deficiency: antigen degradation, endosome acidification, endosome maturation, reduction in MHC-I recycling, antigen trafficking towards lysosomes and tubulation scission. But which of these effects now are responsible for impaired antigen presentation?

To explain the Ag cross-presentation defect evidenced in our mouse model, different hypotheses can be proposed. First, the drastic defect in Ag degradation, that impairs the generation of proteolytic peptides derived from Ag, could be the main cause responsible to impaired Ag cross-presentation. Even though, it is now well established that increased Ag degradation has a detrimental effect on efficient cross-presentation, it is possible that a profound impairment of Ag degradation will impact cross-presentation. Indeed, this is illustrated by the effect of removing the cysteine protease cathepsin S that significantly affects cross-presentation of some Ag forms *in vitro* and *in vivo* (Shen Lj et al. Immunity 2004). Alternatively, the Ag cross-presentation defect can be mainly related to defective endosomal and MHC-I trafficking and recycling. The lack of MHC-II Ag presentation defect will support this hypothesis as the Ag degradation observed in Kif5b-deficient DCs has no detectable effect on the MHC-II Ag presentation pathway. A third hypothesis could be the additive effect of defective Ag degradation and MHC-I recycling in impaired Ag cross presentation. We have now discussed these possibilities in the revised discussion of our manuscript, pages 13-14.

As the authors point out correctly in their discussion, it is generally accepted that rapid antigen degradation prevents cross-presentation, meaning that impaired antigen degradation and trafficking towards lysosomes might not be critically involved in inhibition of cross-presentation.

We do not completely agree with the reviewer's comment that "impaired antigen degradation and trafficking towards lysosomes might not be critically involved in inhibition of cross-presentation". As we proposed above, even if an efficient cross-presentation has been correlated with a reduction in antigen degradation, we cannot exclude that also a profound impairment of Ag degradation could also lead to impair cross-presentation. Indeed, this is illustrated by the effect of removing the cysteine protease cathepsin S that considerably affects cross-presentation (Shen Lj et al. Immunity 2004).

Alternatively, if peptides derived from soluble antigens are loaded onto MHC I in endosomes (regardless of vacuolar or endosome-to-cytosol pathway), they need to be transported towards the cell membrane, which presumably occurs via Rab11+ endosomes. Since the authors show that Kif5b deficiency prevents such protein transport, this might be the main reason for impaired cross-presentation. The discussion on how Kif5b might influence cross-presentation should be extended in the manuscript in general.

As proposed by the reviewer, we have now extended the discussion on how Kif5b might influence cross-presentation (Pages 13-14).

In response to Reviewer #3

This manuscript investigates kinesin-1 in cross-presentation. This is not the first time that kinesin-1 has been reported to play a role in DC function and cross-presentation (this should be cited in the Introduction) and/or receptor recycling.

We do not agree with the reviewer's statement. To our knowledge, our manuscript is the first investigating the role of kinesin-1 in DC function and cross-presentation. Can the reviewer give the complete references of the papers to which he or she refers?

Here, the authors investigate the response in more detail. Overall the manuscript lacks rigour and coherence.

We do not agree with this statement, as do Reviewer 2 who stated that our study is "very clear and describes a novel role of Kif5b in cross-presentation, which is an interesting subject to the field".

A detailed response to each of the reviewer's comments is provided below.

A major shortfall of the manuscript is that there is an overinterpretation of a specific role for kinesin-1 in cross-presentation.

We thank the reviewer for his comment. We have reformulated the text accordingly.

The manuscript lacks in depth analysis of other forms of antigen presentation.

The manuscript mainly investigates soluble OVA as a model antigen. However, we also present additional data regarding defective antigen degradation and cross-presentation to particulate antigen (OVA coupled to latex beads) as supplementary data (Fig. S3 and Fig.S8).

The role of kinesin-1 in MHC II antigen presentation and direct MHC I antigen presentation needs to be clarified in a more rigorous manner. Indeed it is surprising that MHC II antigen presentation is not affected given the significant impact of kinesin-1 KO on endosomal biology and antigen proteolysis.

As mentioned above in the response to the reviewer 1, we have studied the invariant chain Ii degradation to further characterize a role of Kif5b in MHC-II antigen presentation, as Ii is well known to have a key role in MHC-II trafficking and peptide loading. WT and cKO^{Kif5b} BMDCs were analysed by western blotting for the presence of Ii and its intermediate degradation products. The Ii degradation was not found altered in Kif5b-deficient BMDCs. This result is now added to Fig. S5B and Page 5-6.

We then investigated the *in vivo* proliferation of injected OT-II CD4⁺ T cell (labelled by violet proliferation dye) in WT and cKO^{Kif5b} mice that were primed the next day with fusion protein containing OVA (P3UOVA), complexed with hamster anti-mouse CD11c antibody (CD11c/P3UOVA) to target specifically CD11c⁺ DCs (Kratzer R et al. JI 2010). Confirming our previous *in vitro* data (Fig.S5A), we did not observe different proliferation ratio between OT-II CD4⁺ T cells injected in WT or cKO^{Kif5b} mice. These new results are now included as supplemental data in Figure S5C and in the text Page 6 to strengthen that Kif5b does not play a role in MHC-II antigen presentation.

To explain the absence of Kif5b detectable role in MHC-II antigen presentation, we can speculate that the Ag degradation defect (which is not totally but only partially impaired and delayed) in Kif5b-deficient DCs does not affect with the same severity the MHC-II pathway. Furthermore, we cannot rule out the possibility that other member(s) of the kinesin family may regulate MHC-II trafficking or compensate for the loss of kinesin-1. Finally, we cannot exclude

that the Ag degradation defect will not be directly responsible for the Ag cross-presentation defect and thus will not affect MHC-II Ag presentation. Indeed, it is possible that the Ag cross-presentation defect mainly depends of the defective MHC-I trafficking and recycling and thus will not impact MHC-II presentation. We have now discussed this point in the P14 of the discussion section of the revised manuscript.

To assess kinesin-1's role in direct presentation, BMDCs from WT or cKO^{Kif5b} mice were infected by the vaccinia virus-encoded OVA epitope and cultured with OT-I T cells. Secretion of IFN- γ by OT-I cells was assessed (Fig.S4A). Formation of S8L/H2-Kb complexes at the cell surface of BMDC WT and cKO^{Kif5b} evaluated 6 h after infection with vaccinia viruses expressing OVA was assessed by staining cells with the 25-D1.16 monoclonal antibody (Fig.S4B). Our data show that the direct presentation of intracellular OVA was not impaired in Kif5b-deficient BMDCs.

Likewise, direct MHCI presentation may be altered due to the role of kinesin-1 in MHCI recycling. The supplementary data in SFig4 suggests there is a defect in MHCII presentation of soluble OVA by CD8 DC.

In supplemental Fig4 the difference is not significant. This information is now added to the corresponding figure legend.

Moreover, the manuscript requires clarification regarding the different forms of antigen used. Bead-OVA antigen presentation data should be included.

We present in supplementary data (Supplemental Fig3 and Supplemental Fig8) a defect of cross-presentation and antigen degradation to particulate antigen (OVA coupled to latex beads). For the sake of readability of Fig1 and Fig3, we choose to show these data as supplementary material. However, if the editor estimates that these data should be better highlighted, we could still include them in the main Figures.

The data for cross-presentation contradicts the accepted view that cDC1 are the more efficient cross-presenting cell type. For example, published data suggests there is little cross-presentation of soluble OVA by CD11b+ DC eg. Schnorrer. 2006. How do the authors explain this discrepancy between their cross-presentation outcomes and those previously reported?

It is indeed well accepted that CD8+ conventional DCs (cDC1) are specialized in capturing dying cells and cross-present cellular antigens *in vivo*. Nevertheless, several studies show that *in vitro*, CD8- conventional DCs can also cross present with the same efficiency that CD8+ conventional DCs, when incubated with a high concentrations of soluble OVA or OVA-beads (Kamphorst et al. JI 2010, den Haan et al. J.exp.Med. 2002, Iyoda et al. J.exp.Med. 2002).

The link between cross-presentation of soluble antigen *in vitro* (Fig 1) and cross-presentation of tumour associated antigen *in vivo* (Fig 2) is not clear, given these are two very different types of antigen.

To clarify this point and show defect of cross-presentation *in vivo*, we have performed a new set of experiments that assess OT-I T cell proliferation in WT and cKO^{Kif5b} mice that were primed the next day with fusion protein containing OVA (P3UOVA), complexed with hamster anti-mouse CD11c antibody (CD11c/P3UOVA) to target specifically CD11c⁺ DCs (Kratzer R et al. JI 2010). These data are now shown in the revised Fig 2C and in the text Page 6. They indicate a lower rate of proliferation of OT-I T cells in the spleen of cKO^{Kif5b} mice compared to WT mice, thus confirming the inability of Kif5b-deficient DCs to cross-present *in vivo*.

Tumour data is used as evidence for a role of kinesin-1 in cross-presentation *in vivo*. While there is clearly an impact of kinesin-1 on the growth kinetics of the tumour, this also occurs in

the absence of OT-I transfer suggesting mechanisms other than OVA cross-presentation could be playing a role. Are other immune populations eg CD8 and CD4 T cells, B cells and NK cells functional and present in normal numbers in the kinesin-1 cKO?

The cKO^{Kif5b} mice display no obvious abnormality in the lymphoid and myeloid lineage counts (Munoz et al. JCB 2016). We have now added in Fig.S1 the immune phenotype of the lymphoid lineage of cKO^{Kif5b} mice compared to WT mice.

We have also looked at the immune population 10 days after the tumour injection. No difference in the number of CD8+ T cells, CD4+ T cells, B cells and NK cells was detected in cKO^{Kif5b} as compared to WT mice. We have now added these results as supplementary data figure S6A and Page 6. Moreover, *in vitro* assays revealed that NK cKO^{Kif5b} have similar degranulation capacity than WT cells, suggesting that cytotoxic NK cell function was not impaired by the absence of Kif5b. These new results were added in Fig. S6B and Page 6.

Concluding the changes in tumour growth and/or mouse survival are solely due to cross-presentation is an overinterpretation of the data. To measure cross-presentation *in vivo*, the authors need to perform *in vivo* transfer of OTI and OTII cells and measure proliferation in response to different forms of (cross-presented) antigen.

As suggested by the reviewer we have performed a new set of experiments to assess a possible defect of cross-presentation *in vivo* in cKO^{Kif5b} mice. To this end, we investigated the *in vivo* proliferation of OT-I and OT-II T cells (labelled by Violet proliferation dye) in WT and cKO^{Kif5b} mice that were primed the next day with fusion protein containing OVA (P3UOVA), complexed with hamster anti-mouse CD11c antibody (CD11c/P3UOVA), to target specifically CD11c⁺ DCs (Kratzer R et al. JI 2010). As discussed above, we identified a lower rate of proliferation of OT-I T cells in the spleen of cKO^{Kif5b} mice compared to WT mice, confirming the inability of Kif5b-deficient DCs to cross-present *in vivo*. In contrast, proliferation of OT-II cells was similar in WT and cKO^{Kif5b} mice, confirming that MHC-II presentation was not affected in DCs lacking Kif5b. We have now added these results in Fig. 2C and in Fig. S5C.

In Figure 5, analysis of OVA trafficking shows higher colocalization with EEA-1 and less with LAMP and Rab11 early in the cKO but at later time points this is enhanced compared to WT. How do the authors explain this? Do these experiments take into account that in the WT there is significantly more antigen proteolysis compared to cKO?

As mentioned above in response to reviewers 1 and 2, we explain this discrepancy by the impairment of soluble ovalbumin degradation observed in Kif5b-deficient BMDCs compared to WT BMDCs (Fig3B) and the delay of recruitment of early endosomes to late endosomes/lysosomes also observed in Kif5b-deficient BMDCs compared to WT BMDCs (Fig6H, 6I). We have clarified this point in the revised version of the manuscript Page 8.

It is unclear how the data in Figure 7 directly relates to kinesin-1. Given that tubule formation requires actin polymerisation then it is not surprising endosomal tubulations would not be present in cells without tubules. It is unclear how this data fits into the theme of the manuscript.

In Figure 7, we have tried to decipher the molecular mechanism required to promote nascent tubule formation, tubule elongation and tubulation scission of the early endosome in DCs. We have shown that actin cytoskeleton promotes nascent tubule formation and elongation whereas microtubules are required only for tubulation scission. Interestingly, disruption of the microtubules by nocodazole mimics the defective function of Kif5b in BMDCs. Fig7 illustrates the different roles of Kif5b on actin cytoskeleton and microtubules in this process that were not so predictable as members of the kinesin superfamily were previously described in other systems to participate to the tubule elongation (Roux et al. 2002 PNAS, Delevoye et al. 2014 Cell Rep).

The Discussion is poorly written and needs to be more concise.
We have now rewritten the discussion, trying to be more concise.

Throughout the manuscript there are many statements that over-interpret the data presented:
"Our results provide the first genetically based evidence in support of this hypothesis, and highlight kinesin-1 as a critical regulator of Ag degradation and Ag sorting from the early endosomes."

"Our results highlight kinesin-1's newly recognized role as a molecular checkpoint that modulates the balance between antigen degradation and cross-presentation."
"Our results show that kinesin-1 (i) has an essential role in the Ag and MHC-I endocytic trafficking upstream of cross-presentation"

"Taken as a whole, these results suggest that the defect in T cell priming observed in the absence of kinesin-1 was specific to cross-presentation." "In contrast, the presentation of endogenous Ag to MHC-I molecules and the presentation of exogenous Ag to MHC-II molecules in cKOKif5b mice was unaffected".

"Taken as a whole, these data indicate that Kif5b binds to EEAI-positive sorting endosomes, drives the scission of tubular structures, enables vesicles to mature into recycling or late endosomes, and thus allows the subsequent sorting of Ag/MHC-I complexes"

We have now reformulated the text to avoid overinterpretation of our data and to make the discussion more concise.

The majority of the statistical analysis of data is performed using an unpaired t test which is not the correct analysis for the type of data presented.

Following reviewer's suggestion, we have now used one-way ANOVA (one independent variable in our analysis of variance test) to compare the means of three groups for Fig12S, two-way ANOVA (two independent variables) for Fig1D, E, F; Fig 2A; Fig3B, C, D; Fig4; Fig5B, C, D; Fig6G, H; Fig7B; FigS3 and FigS8 and unpaired t-test was still used to compare the means of two groups for Fig2C; Fig5E and Fig6F. This statistical analysis confirmed the previous ones.

Minor Comments

More detail is required to explain Fig 1A in the text, how was Kif5b measured, what are Kif5c, Kif5a?

As proposed by the reviewer, we have now added more explanation for Fig1A in the revised text of the manuscript Page 5.

Fig 2C. Mouse tumour images are unnecessary.

Following reviewer's suggestion, we decided not to include Fig2C in the revised version of the manuscript. However, if the editor estimates that these data are needed, we could still include them in Fig2.

Figure panels for Figure 3 should be ordered so CD8 DC are aligned.

In the revised version of Fig3, we have respected the same order for the different DC-types.

Have technical repeats been performed for endosomal pH measurements?

The data are representative of four independent experiments. We have now added statistical analysis for the pH experiment (new figure 3D).

In response to Reviewer #4

The authors show here that in absence of Kif5b expression, cross presentation in DCs is impaired and anti tumor immune responses are reduced. They also show that endocytic degradation, acidification, recycling and cargo transfer to late compartments are all impaired.

The results are clearcut and the experience are well designed. The authors conclude that these defects in intracellular traffic cause reduced antigen cross presentation and impair anti tumor immune responses. They propose that Kif5b plays a critical role in the control of the balance between antigen degradation and cross presentation.

The main problem with this conclusion is that, based on previous studies (cited by the authors), the defects in endocytic functions and intracellular traffic in Kif5b KO dendritic cells would suggest increased and not impaired, antigen cross presentation. Several groups have shown that reduced degradation, acidification and lysosomal transport are all associated with better cross presentation. The main difference with the Kif5b KO is that recycling is also impaired. It is therefore most likely that impaired cross presentation in Kif5b is in fact due, mainly, to defective MHC I re-cycling. Supporting this possibility with experimental data would strengthen the paper. And in all cases this issue should be better discussed.

As mentioned above in the response to reviewer 2, even though, it is now well established that increased Ag degradation has a detrimental effect on efficient cross-presentation, it is possible that a profound impairment of Ag degradation will impact cross-presentation. Indeed, this is illustrated by the effect of removing the cysteine protease cathepsin S that significantly affects cross-presentation of some Ag forms *in vitro* and *in vivo* (Shen Lj et al. Immunity 2004). Alternatively, the Ag cross-presentation defect can be mainly related to defective endosomal and MHC-I trafficking and recycling. The lack of MHC-II Ag presentation defect will support this hypothesis as the Ag degradation observed in Kif5b-deficient DCs have no detectable effect on the MHC-II Ag presentation pathway. A third hypothesis could be the additive effect of defective Ag degradation and MHC-I recycling in impaired Ag cross presentation. We have now discussed this point in the revised discussion pages 13-14.

What are the effects of nocodazole and latruculin B on MHC class I re-cycling?

We did not assess the effects of nocodazole and latruculin B on MHC class I re-cycling, as the results will not be related to kinesin-1 function.

Other remarks:

- In the tumor growth experiment it is important to monitor the T cell responses (eventually using a model tumor antigen).

As suggested by the reviewer, we have monitored T cell responses *in vivo* in cKO^{Kif5b} mice. To this end, we investigated the *in vivo* proliferation of OT-I and OT-II T cells (labelled by violet proliferation dye) in WT and cKO^{Kif5b} mice that were primed the next day with fusion protein containing OVA (P3UOVA), complexed with hamster anti-mouse CD11c antibody (CD11c/P3UOVA), to target specifically CD11c⁺ DCs (Kratzer R et al. JI 2010). As discussed above, we identified a lower rate of proliferation of OT-I T cells in the spleen of cKO^{Kif5b} mice compared to WT mice, confirming the inability of Kif5b-deficient DCs to cross-present *in vivo*. In contrast, proliferation of OT-II cells was similar in WT and cKO^{Kif5b} mice, confirming that MHC-II presentation was not affected in DCs lacking Kif5b. We have now added these results in Fig. 2C and in Fig. S5C.

We have also looked at the immune populations 10 days after the tumour injection. No difference in the number of CD8⁺ T cells, CD4⁺ T cells, B cells and NK cells was detected in cKO^{Kif5b} as compared to WT mice. We have now added these results as supplementary data

figure S6A and Page 6. Moreover, *in vitro* assays revealed that NK cKO^{Kif5b} have a similar degranulation capacity than WT cells, suggesting that cytotoxic NK cell function was not impaired by the absence of Kif5b. These new results were added in Fig. S6B and Page 6.

- In all figures showing different types of DCs, please keep the same order for the different DC-types.

In the revised version of Fig3 and Fig5, we have respected the same order for the different DC-types.

- The summary scheme in Figure 7F is not very helpful... If kinesin 1 (does this means Kif5b?) is involved in both recycling and degradation (as the results show), it is not possible to know if the effect on cross presentation is due to either or both...

As figure 7F does not seem to fit with the message of the manuscript, we have decided not to include Fig7F in the revised version of the manuscript. We made a new summary scheme in supplemental figure 13.

REVIEWERS' COMMENTS:

Reviewer #1 (Remarks to the Author):

The authors have answered most of my questions thus I recommend the paper for publication.

Reviewer #2 (Remarks to the Author):

The authors have addressed my main concerns.

Reviewer #3 (Remarks to the Author):

The control experiments examining MHC II antigen presentation are not performed under the same conditions as those examining cross-presentation and therefore it remains difficult to draw rigorous conclusions from Figure 1 that the defect is cross-presentation specific. Figure 1 is currently a mismatch of different types of antigens tested and assays performed to evaluate the different antigen presentation pathways. The authors show Kif5bKO DC have reduced cross-presentation of soluble antigen and latex bead-antigen in vitro. The MHC II antigen presentation assay is performed by priming mice in vivo with anti-CD11c-OVA and monitoring division of transferred T cells. These assays need to be aligned ie MHC I and MHC II antigen presentation evaluated under same conditions in the same type of assay.

Supplementary data S3, S4 and S5A should be in manuscript to more rigorously conclude this is a specific defect in cross-presentation.

Weimershaus (ref 32) should be discussed in more depth regarding how kinensin 3 participates in cross-presentation and how this differs from proposed role of kinensin 1.

Minor

"responded well to lipopolysaccharide"- should be edited, not clear what "well" means

Fig1F: Were the GMCSF BMDC cultures similar for WT and Kif5b KO? How these DCs were sorted (ie what markers) for the cross-presentation assay should be mentioned in Methods.

Reviewer #4 (Remarks to the Author):

The authors have addressed my concerns.

Reviewer #1 (Remarks to the Author):

The authors have answered most of my questions thus I recommend the paper for publication.

Reviewer #2 (Remarks to the Author):

The authors have addressed my main concerns.

Reviewer #4 (Remarks to the Author):

The authors have addressed my concerns.

Reviewer #3 (Remarks to the Author):

The control experiments examining MHC II antigen presentation are not performed under the same conditions as those examining cross-presentation and therefore it remains difficult to draw rigorous conclusions from Figure 1 that the defect is cross-presentation specific. Figure 1 is currently a mismatch of different types of antigens tested and assays performed to evaluate the different antigen presentation pathways.

Figure 1 only investigates soluble Ovalbumin (sOVA) as a model antigen. However, we also present additional data regarding defective antigen degradation and cross-presentation to particulate antigen (OVA coupled to latex beads) as supplementary data (Fig. S3 and Fig.S8).

The authors show Kif5bKO DC have reduced cross-presentation of soluble antigen and latex bead-antigen *in vitro*. The MHC II antigen presentation assay is performed by priming mice *in vivo* with anti-CD11c-OVA and monitoring division of transferred T cells. These assays need to be aligned ie MHC I and MHC II antigen presentation evaluated under same conditions in the same type of assay.

In order to evaluate the role of Kif5b in MHC-II antigen presentation, we have performed:

- MHC-II presentation of sOVA in Kif5b-deficient DCs (CD8 α ⁺ and CD11b⁺ DCs and BMDCs) (as probed by assaying IL2 production by OT-II T cells) (Supplementary Fig. 5a)
- we have studied the invariant chain Ii degradation (Supplementary Fig. 5b)
- we have investigated the *in vivo* proliferation of injected OT-II CD4⁺ T cell in WT and cKO^{Kif5b} mice that were primed the next day with fusion protein containing OVA (P3UOVA), complexed with hamster anti-mouse CD11c antibody (CD11c/P3UOVA) to target specifically CD11c⁺ DCs. (Supplementary Fig. 5c)

These three different approaches did not evidence a role of kinesin-1 in MHC-II antigen presentation. We could not use sOVA as antigen *in vivo* due to a lack of specificity.

Supplementary data S3, S4 and S5A should be in manuscript to more rigorously conclude this is a specific defect in cross-presentation.

For the sake of readability of Fig1 and Fig3, we choose to show these data as supplementary material. However, if the editor estimates that these data should be better highlighted, we could still include them in the main Figures.

Weimershaus (ref 32) should be discussed in more depth regarding how kinensin 3 participates in cross-presentation and how this differs from proposed role of kinensin 1.

The reviewer has raised a very interesting point. However, it is not so easy to have a clear view of the role of kinesin-3 (KIF16B) in our model. Kinesin-3 seems to interact with Rab14 which label specific subset of IRAP⁺ endosomes and not EEAI-containing endosomes. Kif16B

depletion has been shown to abolish recruitment of Rab14 to phagosomes and accelerated their maturation. KIF16b knock-down reduced cross-presentation of phagocytized Ag by BMDCs by a probable excessive degradation of Ag. This suggest that kinesin-1 and kinesin-3 have opposite function and do not regulate the same endosomes.

Minor

"responded well to lipopolysaccharide"- should be edited, not clear what "well" means
As suggested by the reviewer, we have edited "well" by "normally".

Fig1F: Were the GMCSF BMDC cultures similar for WT and Kif5b KO? How these DCs were sorted (ie what markers) for the cross-presentation assay should be mentioned in Methods.

For the generation of BMDCs, bone marrow was isolated from femurs of 8- to 12-week-old mice. Cells were then cultured in medium supplemented with GM-CSF, 15% heat-inactivated FCS, 1% non-essential amino acids, 1 mM sodium pyruvate, 100 U/ml penicillin, 100 U/ml streptomycin. After 8 to 10 days of culture, the BMDCs were used in experiments. No differences were observed between WT and cKO^{Kif5b} BMDCs (Supplementary Fig. 2). For cross-presentation BMDCs are not sorted, only secretion of IL-2 was quantified using an ELISA to monitor OT-I activation reflecting the ability of BMDC to cross-present.